# Angle Graph Transformer: Capturing Higher-Order Structures for Accurate Molecular Geometry Learning

## Abstract

Existing Graph Transformer models primarily focus on leveraging atomic and chemical bond properties along with basic geometric structures to learn representations of fundamental elements in molecular graphs, such as nodes and edges. However, higher-order structures like bond angles and torsion angles, which significantly influence key molecular properties, have not received sufficient attention. This oversight leads to inadequate geometric conformation accuracy and difficulties in precise local chirality determination, thereby limiting model performance in molecular property prediction tasks. To address this issue, we propose the Angle Graph Transformer (AGT). AGT directly models directed bond angles and torsion angles, introducing higher-order structural representations to molecular graph learning for the first time. This approach enables AGT to determine local chirality within molecular representations and directly predict torsion angles. We introduce a novel Directed Cycle Angle Loss, allowing AGT to predict bond angles and torsion angles from low-precision molecular conformations. These properties, along with interatomic distances, are then applied to downstream molecular property prediction tasks using a pre-trained AGT with Hierarchical Virtual Nodes. Our model achieves new state-of-the-art (SOTA) results on the PCQM4Mv2 and OC20 IS2RE datasets. Through transfer learning, AGT also demonstrates competitive performance on molecular property prediction benchmarks including QM9, MOLPCBA, LIT-PCBA, and MoleculeNet. Further ablation studies reveal that the conformations generated by AGT are closest to conformations generated by Density Functional Theory (DFT) among the existing methods, due to the constraints imposed by the bond angles and torsion angles.

## 1 Introduction

Transformer (Vaswani, 2017) models have expanded from natural language processing to various domains (Dosovitskiy, 2020; Child et al., 2019). Due to their ability to capture long-range dependencies between nodes, Transformers have been widely applied to graph data. Graph Transformers (GTs) (Ying et al., 2021; Hussain et al., 2022; Feng et al., 2022; Zhou et al., 2023) have demonstrated potential surpassing message-passing neural networks on diverse graph datasets, including superpixels, citation networks, and molecular graphs. Following the trend of model scaling in various domains (Brown et al., 2020; Chowdhery et al., 2022; Borgeaud et al., 2022), increasing model capacity through well-designed architectures has shown improved information capture and stronger generalization capabilities in downstream tasks. Building upon this foundation, the Alphafold (Jumper et al., 2021) series of works emerged, achieving remarkable results in protein structure prediction and propelling life science research forward in a leap-like manner.

Most Graph Transformers primarily use nodes as tokens, employing global attention to facilitate information exchange across the entire graph. In the domain of molecular graph data, 3D structural information of molecules is often closely related to molecular properties and is thus typically encoded in the model and trained as a key attribute (Zhou et al., 2023; Stärk et al., 2022). The EGT (Hussain et al., 2022) introduces edge embeddings as tokens, enabling new pairwise information to be updated through dedicated channels in consecutive layers. Recently, researchers have noted the performance improvements achieved by the triangle inequality constrained interatomic

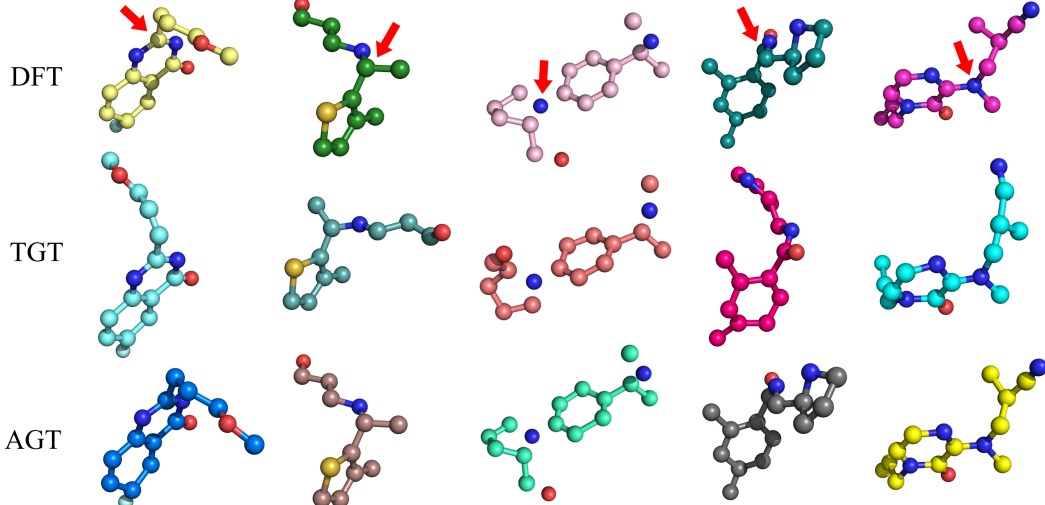

Figure 1: The ability to identify local chirality. The first row depicts DFT conformations. The second and third rows show the corresponding molecular conformations from TGT distance predictor and AGT conformations predictor. AGT can accurately generate molecules with local chirality identical to the target conformation, whereas TGT conformations, relying solely on distance matrices, exhibit deviations. Red arrows indicate atoms representing the centers of local chirality in the molecules.

distance prediction method in AlphaFold (Jumper et al., 2021). Consequently, they proposed Uni-Mol+ (Lu et al., 2023) and TGT (Hussain et al., 2024), both utilizing axial attention to satisfy the communication pattern where three pairwise relationships in a triangle are interconnected. These method overcomes the information exchange bottleneck, allowing edge embeddings to better adhere to geometric constraints when predicting distances.

Although this triangular inequality constraint can optimize the geometric spatial structure of predicted conformations, two significant issues remain unresolved. Firstly, as described in AlphaFold 3 (Abramson et al., 2024), merely predicting interatomic distances is insufficient to determine the local chirality of geometric conformations. Local chirality refers to the inability of a specific part or group within a molecule to superimpose on its mirror image through central symmetry rotation. Local chirality is crucial for the functionality of many biomolecules, such as the active sites of enzymes. However, Molecules with different local chirality may yield similar distance matrices, especially in small molecule conformations, and may even produce identical distance matrices. This limitation makes it impossible to determine the local chirality of generated conformations, increasing the ambiguity in molecular representation. Secondly, conformations generated solely based on distance matrices tend to exhibit instability in predicting torsion angles. Existing GT architectures do not treat the torsion angle as a unified higher-order graph substructure, resulting in each torsion angle being constructed from three separate pairwise embeddings. Consequently, small errors in each distance prediction can accumulate multiplicatively in the torsion angle, leading to significant deviations in the generated conformation's torsion angles. This can cause changes in the overall molecular conformation, affecting the prediction of molecular function.

To address these two major challenges, we propose the Angle Graph Transformer (AGT), a model that directly models higher-order graph substructure representations such as bond angles and torsion angles. AGT treats bond angles and torsion angles as individual tokens in the self-attention mechanism for direct communication, rather than aggregating node and edge representations involved in angles as the final angle representation. This approach of directly interacting at higher-order substructures enables effective global information utilization for predicting torsion angles, overcoming the bottleneck of local information exchange in graph structures and better learning geometric constraints of molecular conformations. To address the inability of existing models to distinguish local molecular chirality, AGT predicts all angles in the range of $(0, 2\pi)$, giving the predicted angles directionality in three-dimensional space. This angular information allows the model to distinguish arbitrary local chirality information in molecules. Additionally, we introduce a hierarchical virtual node aggregation architecture, enabling AGT to directly aggregate information from graph substructures of different orders for prediction.

Based on these contributions, our proposed AGT model surpasses the TGT model on quantum chemistry datasets including PCQM4Mv2, OC20 IS2RE, and QM9, achieving new state-of-the-art re-

sults. We also demonstrate effectiveness of AGT in transfer learning, achieving new SOTA results on molecular property prediction datasets MOLPCBA, MOLHIV, and the drug discovery dataset LIT-PCBA benchmark. This indicates that the geometric features extracted by our trained conformations predictor can be applied to new downstream molecular graph tasks. Results from ablation studies indicate that AGT-generated conformations have discriminative ability in local chirality and are more accurate.

## 2 RELATED WORK

### 2.1 ANGLE PREDICTION IN MOLECULAR CONFORMATION OPTIMIZATION

The incorporation of angular constraints, including bond angles and torsion angles, in molecular conformations has been progressively applied in recent works. GEOMOL (Ganea et al., 2021) was among the earlier methods to introduce torsion angle constraints in three-dimensional conformation generation. TorsionNet (Rai et al., 2022) employed deep neural networks to predict torsional energy distributions of small molecules with quantum mechanical-level accuracy. Subsequently, Torsional diffusion (Jing et al., 2022) proposed a diffusion model framework operating in the torsion angle space. DiffPack (Zhang et al., 2024) learned the joint distribution of side-chain torsion angles by diffusing and denoising in the protein side-chain torsion angle space, while Tora3D (Zhang et al., 2023) predicted a set of torsion angles for rotatable bonds using an interpretable autoregressive method and reconstructed 3D conformations using energy guidance. AUTODIFF (Li et al., 2024a) designed a molecular assembly strategy called conformational motifs to mitigate issues with skewed bond or torsion angles. Our method draws inspiration from the aforementioned works, incorporating angular constraints as a crucial component in rationalizing conformation generation. Notably, while existing works have utilized angular information, they have not addressed the ability to discriminate local chirality. AGT is the first to achieve this using angular information.

### 2.2 PREDICTIVE MOLECULAR STRUCTURAL PRE-TRAINING

AlphaFold (Jumper et al., 2021) employs a Transformer architecture for predictive structural pre-training on vast protein datasets. In the analogous field of small molecule structural pre-training, models based on Graph Transformers (GTs) are at the forefront of research. Previous works such as GraphTrans (Wu et al., 2021), GSA (Rashedi et al., 2009), GROVER (Rong et al., 2020), and GPS (Rampášek et al., 2022) utilized hybrid approaches combining Transformers and Graph Neural Networks (GNNs) to enhance model expressiveness. In contrast, pure GTs instead directly inputting nodes or substructures as tokens into the Transformer for training. The two most representative architectures in this category are exemplified by Graphormer (Ying et al., 2021; Shi et al., 2022) and EGT (Hussain et al., 2022). Graphormer-type models primarily use atoms as tokens, implicitly encoding chemical bond and spatial structure information as additional atom embeddings through positional encoding and attention bias. Notable works in this category include Unimol (Zhou et al., 2023), GEM-2 (Liu et al., 2022a), and Transformer-M (Luo et al., 2022). The other category, represented by the EGT backbone model, is characterized by direct modeling of edges. These models treat edge embeddings as Transformer tokens and employ global attention for information exchange between node and edge tokens. All three aforementioned approaches have seen the emergence of works applying triangular inequality attention, such as GPS++ (Masters et al., 2022), Unimol+ (Lu et al., 2023), and TGT (Hussain et al., 2024). This distance constraint can be equivalently regarded as the interaction of specific axial edge markers in attention. While these methods have achieved excellent performance, they remain limited to edges, the simplest second-order substructure in graphs (composed of two nodes and the connection between them). Naturally, we consider constructing tokens on higher-order substructures (such as third-order bond angles and fourth-order torsion angles) and using attention mechanisms for communication.

## 3 METHOD

AGT initially obtains low-precision 3D conformations using cost-effective methods, i.e., RDKit. Subsequently, it employs a conformer predictor to learn target conformations from these low-precision structures such as high-precision equilibrium conformations optimized through DFT. Finally, the learned conformations are input into the task predictor to forecast molecular properties.

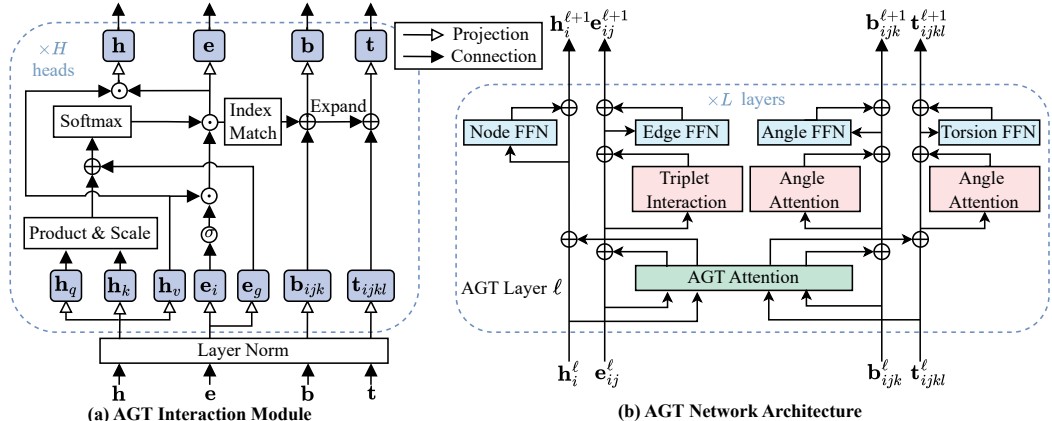

Figure 2: (a) AGT Interaction Module. Index Match denotes the selection of corresponding edge embeddings based on the indices of nodes where angle substructures are located. Expand refers to the dimension augmentation to accommodate torsion angle indices. (b) AGT Network Architecture. Angle Attention take angle substructures as tokens and uses multi-head self-attention mechanism to update the representation.

The overall training process closely resembles that of TGT. While TGT models direct communication between two pairwise elements through triangular inequality attention mechanisms, it lacks modeling of higher-order substructures and cannot accurately discriminate local chirality and angles in the geometric conformation space. AGT addresses these limitations of TGT's edge-only modeling by introducing modeling of higher-order substructures, specifically bond angles and torsion angles. This enhancement enables AGT to achieve greater expressive power.

### 3.1 AGT ARCHITECTURE

The AGT model can be denoted as $(y, \hat{D}, \hat{B}, \hat{T}) = f(X, E, D, B, T; \theta)$. The AGT model utilizes atomic features ($X \in \mathbb{R}^{n \times d_x}$, where $n$ is the number of atoms and $d_x$ is the atom feature dimension), edge features ($E \in \mathbb{R}^{n \times n \times d_e}$, where $d_e$ is the edge feature dimension), and 3D conformational information including the complete distance matrix ($D \in \mathbb{R}^{n \times n}$), all bond angles ($B \in \mathbb{R}^{n_b}$, $n_b$ is the number of bond angles), and torsion angles ($T \in \mathbb{R}^{n_t}$, $n_t$ is the number of torsion angles) within the molecule to predict molecular properties $y$ and update 3D conformational information using learnable parameters $\theta$. The model has $L$ blocks, with $h^{(l)}$, $e^{(l)}$, $b^{(l)}$ and $t^{(l)}$ representing the $l$-th block's outputs.

**The Initialization of Substructure** Atom representations are composed of the atom's inherent properties, while edge representations are formed by the chemical bond properties, the types of atoms at both ends, and the bond length. We opted against modeling substructures using arbitrary combinations of three and four nodes for two reasons. Firstly, unconstrained interactions among triplets and quadruplets would escalate the computational complexity to $O(N^5)$, which is prohibitive for any graph. Secondly, randomly modeled substructures often lack physical significance. Therefore, we adopted an approach that considers only substructures with actual significance in AGT. We identified nodes of triplets and quadruplets connected by consecutive chemical bonds, which correspond to bond angles and torsion angles as higher-order substructures. This approach ensures that substructure features are closely tied to chemical bonds, significantly influencing molecular properties. Simultaneously, the number of higher-order substructures obtained is substantially smaller than the total number of triplets and quadruplets in the complete graph. Consequently, the additional computational complexity introduced in the model generally does not exceed $O(N^2)$.

**AGT Interaction Module** We have redesigned the information interaction mechanism between substructures of different orders, resulting in structural representations that satisfy angular constraints. First, we compute the axial attention for each of the two edges independently. Subsequently, the bond angle embedding is obtained by using the indices of the two edges forming the angle to locate the corresponding positions and summing the embeddings. Similarly, for dihedral angle updates, we use the indices of three consecutive edges that form the torsion angle to locate and sum the corresponding torsion angle embeddings. This approach allows for a hierarchical update of representations of different structural levels in the graph, progressing from atoms to chemical bonds, then

to bond angles, and finally to torsion angles. This hierarchical method enables better integration of substructure features that carry chemical significance. The updates of atom and edge representations in the AGT Module are as follows:

$$e^{(l)} = \frac{h^{(l-1)}W_Q^{(l,h)}\left(h^{(l-1)}W_K^{(l,h)}\right)^T}{\sqrt{d_h}} + e^{(l-1)}W_E^{(l,e)}. \tag{1}$$

$$h^{(l)} = \text{softmax}\left(e^{(l)}\right)\sigma(e^{(l-1)}W_G^{(l,e)})h^{(l-1)}W_V^{(l,h)}. \tag{2}$$

where $d_h$ is the head dimension, $\boldsymbol{W}_Q^{(l,h)}, \boldsymbol{W}_K^{(l,h)}, \boldsymbol{W}_V^{(l,h)} \in \mathbb{R}^{d_a \times d_h}, \boldsymbol{W}_E^{(l,e)}, \boldsymbol{W}_G^{(l,e)} \in \mathbb{R}^{d_p \times d_h}$. The representation of bond angles and torsion angles is achieved by adding the corresponding edge representations to their respective indices:

$$b_{ijk}^{(l)} = \sum (ab) \in \{(ij),(jk),(ki)\}e_{ab}^{(l)} + b_{ijkl}^{(l-1)}W_B^{(l,b)}. \tag{3}$$

$$t_{ijkl}^{(l)} = \sum (ab) \in \{(ij),(jk),(kl),(ik),(jl),(il)\}e_{ab}^{(l)} + t_{ijkl}^{(l-1)}W_T^{(l,b)}. \tag{4}$$

where $\boldsymbol{W}_B^{(l,h)} \in \mathbb{R}^{d_b \times d_h}, \boldsymbol{W}_T^{(l,h)} \in \mathbb{R}^{d_t \times d_h}$. Both bond angles and torsion angles utilize the edge representations from the current layer for aggregation, allowing for an efficient use of atomic and edge representations from the previous layer. The method of edge representation aggregation can lead to varying effects, the results of which are presented in the ablation studies. Following the AGT Module, different order substructures are updated using distinct mechanisms. Similar to TGT, atomic representations are updated using an FFN layer, while edge representations are updated through triplet interaction. For bond angles and torsion angles, we employ self-attention layers to update them independently. This approach aims to facilitate direct information exchange among higher-order substructures across the entire molecular graph without relying on atomic or edge representations. These updates can be formulated as follows:

$$\boldsymbol{h}^{(l)} = \boldsymbol{h}^{(l-1)} + \text{FFN}\left(\boldsymbol{h}^{(l)}\right);$$

$$\boldsymbol{e}^{(l)} = \boldsymbol{e}^{(l-1)} + \text{FFN}\left(\text{TripletInteraction}\left(\boldsymbol{e}^{(l)}\right)\right);$$

$$\boldsymbol{b}^{(l)} = \boldsymbol{b}^{(l-1)} + \text{FFN}(\text{softmax}(\frac{W_Q^{(l,b)}\boldsymbol{b}^{(l)}\left(W_K^{(l,b)}\boldsymbol{b}^{(l)}\right)^T}{\sqrt{d_b}})W_V^{(l,b)}b^{(l)}); \tag{5}$$

$$\boldsymbol{t}^{(l)} = \boldsymbol{t}^{(l-1)} + \text{FFN}(\text{softmax}(\frac{W_Q^{(l,t)}\boldsymbol{t}^{(l)}\left(W_K^{(l,t)}\boldsymbol{t}^{(l)}\right)^T}{\sqrt{d_t}})W_V^{(l,t)}t^{(l)}).$$

**Directed Cycle Angle Loss (DCA loss)** AGT extends molecular geometry prediction from full distance matrices to include both bond angles and torsion angles, relying on these angles to determine local molecular chirality. By definition, when local molecular chirality changes, at least one torsion angle or bond angle $\sigma$ will change to $2\pi - \sigma$, given a fixed direction (e.g., counterclockwise). Methods that only predict interatomic distances face significant challenges in determining bond and torsion angles unambiguously. First, both $\sigma$ and $2\pi - \sigma$ can satisfy the same distance matrix in 3D space. Moreover, when local chiral structures are at molecular terminals and other asymmetric structures are distant from the local chiral structures, the differences in distance matrices induced by chirality become extremely subtle, making chirality prediction solely through distance matrices highly sensitive to noise. Previous works on predicting angles often neglected the direction of angles, simply constraining angles to the range of 0 to $\pi$. This limitation results in learned representations that fail to fully capture chiral variations. Another challenge lies in the cyclic nature of angle prediction, which differs from distance prediction. To address these, AGT employs a directed circular binning loss to compute angle loss, more accurately reflecting the proximity between predicted and true values. The specific loss can be expressed as:

$$L_{DCA} = \min\left(-\sum_{i=1}^{N} q_i \log(p_i), -\sum_{i=1}^{N} q_i \log(p_{(i+1) \mod N})\right). \tag{6}$$

Where $q_i$ is ground truth angle distribution, $p_i$ is the predicted angle distribution and $N$ is the number of bins. We extends the angle range to $(0, 2\pi)$ and designates the counterclockwise direction as

Figure 3: The three stages of AGT training.

primary, enabling representation of all local chirality change scenarios. When the prediction is close to $2\pi$ while the true value is near 0 (or vice versa), the shifted distribution will yield a small loss, correctly reflecting the proximity of these two angles. This improvement ensures that the loss function behaves more reasonably when dealing with angles near the boundaries, avoiding excessive penalization of angle values that are actually very close. It also naturally handles cases that cross the $0/2\pi$ boundary.

**Hierarchical Virtual Node** Recent studies (Li et al., 2024b; Xing et al., 2024) have demonstrated that employing virtual nodes in graph data helps mitigate information bottlenecks and over-globalizing issues. Previous research on molecular property prediction was either an aggregated representation of all atoms or the use of atomic level virtual nodes as the final output. However, merging atomic representations often leads to information compression, potentially resulting in the loss of critical structural details and overlooking the contributions of specific structural elements to molecular properties. Using atomic level virtual nodes solely may inadequately represent the complex interactions between atoms in three-dimensional space. To address these limitations and directly capture the impact of substructures, we propose an extended virtual node method in AGT called hierarchical virtual nodes. For each type of substructure, AGT constructs a virtual node to interact with the same type of substructure tokens. Atomic virtual nodes and atom tokens both are trained by the FFN layer; edge virtual nodes participate in normal edge tokens interaction; bond angle virtual nodes undergo self-attention layers with bond angle tokens, and torsion angle virtual nodes follow the same mechanism. Subsequently, for property prediction tasks, we construct a molecule-level virtual node connected to the four substructure virtual nodes, serving as the final output for prediction. We employ hierarchical virtual nodes only during the pre-training phase.

## 3.2 MODEL TRAINING

training procedure of AGT includes three stages for molecular property prediction task. First, in the conformation prediction stage, a conformation predictor is trained to predict the accurate molecular conformations based on low-precision 3D molecular structures. Second, during the pre-training stage, a task predictor is employed to predicts molecular properties from the pre-training dataset. This predictor also receives noisy conformational structures as input and denoise conformational structures. Finally, in the fine-tuning stage, the frozen, pre-trained conformation predictor and task predictor are fine-tuned on downstream datasets.

**Conformer Prediction Stage** We train the AGT conformation predictor to predict all pairwise interatomic distances, bond angles, and torsion angles within a molecule. The conformation predictor takes a low-precision 3D conformation as input (typically an RDKit conformation) and outputs all pairwise interatomic distances, bond angles, and torsion angles. Angles are invariant to translation and rotation, and their values have a fixed range. Inspired by TGT, we predict binned angles instead of continuous values, as torsion angle structures are typically less stable than chemical bonds and more susceptible to rapid changes due to molecular energy fluctuations. The AGT employs cross-entropy loss for pairwise atomic distances and the Directed Cycle Angle Loss for angles.

**Pre-training Stage** In the pre-training phase, AGT train the AGT task predictor on noisy ground truth 3D conformations. This approach ensures that the task predictor is robust to noise in both input

Table 1: Results on PCQM4MV2 valid set.

| Model | # param. | # layers | MAE (meV)↓ |
|---|---|---|---|
| MLP-Fingerprint (Hu et al., 2022) | 16.1M | - | 173.5 |
| GCN (Kipf & Welling, 2016) | 2.0M | - | 137.9 |
| GIN (Xu et al., 2018) | 3.8M | - | 119.5 |
| GINEv2 (Brossard et al., 2020) | 13.2M | - | 116.7 |
| GIN-VN (Xu et al., 2018; Gilmer et al., 2017) | 6.7M | - | 108.3 |
| DeeperGCN-VN (Li et al., 2020) | 25.5M | 12 | 102.1 |
| TokenGT (Kim et al., 2022) | 48.5M | 12 | 91.0 |
| EGT (Hussain et al., 2022) | 89.3M | 18 | 86.9 |
| GRPE (Park et al.) | 46.2M | 18 | 86.7 |
| Graphormer (Ying et al., 2021; Shi et al., 2022) | 47.1M | 12 | 86.4 |
| GraphGPS (Liu et al.) | 13.8M | 16 | 85.2 |
| GEM-2 (Liu et al., 2022a) | 32.1M | 12 | 79.3 |
| GPS++ (Masters et al., 2022) | 44.3M | 16 | 78.1 |
| Transformer-M (Luo et al., 2022) | 69M | 18 | 77.2 |
| Uni-Mol+ (Lu et al., 2023) | 77M | 18 | 69.3 |
| TGT (Hussain et al., 2024) | 203M | 24 | 67.1 |
| AGT | 68M | 6 | 69.4 |
| | 127M | 12 | 69.1 |
| | 241M | 24 | **66.2** |

distances and angles, enabling it to adapt to approximate conformations output by the conformation predictor, which still contain noise and errors. We maintain predictions for pairwise interatomic distances, bond angles, and torsion angles. This auxiliary task encourages different order substructure representations to denoise the 3D structure, optimizing various order substructure representations through self-supervised signals from the molecular structure itself. We combine distance prediction loss and angle prediction loss as secondary objectives with the primary tasks from the pre-training dataset in a multi-task learning framework to jointly train AGT's task predictor. Furthermore, AGT employs hierarchical substructure virtual nodes for joint prediction in molecular property prediction, facilitating the association between substructures and molecular properties.

**Fine-tune Stage** In the fine-tuning phase, AGT employs a frozen, pre-trained conformation predictor to generate DFT conformations from RDKit conformations, thereby obtaining high-precision 3D structural features of molecules. During this process, the conformation predictor operates in stochastic mode with active dropout (Hussain et al., 2024). Subsequently, the predicted bond angles, torsion angles, and distances serve as input to the task predictor. The fine-tuning process combines the primary objective of the downstream dataset's task with auxiliary optimization functions for distance and angle. We utilize the model-generated atomic distance matrix, bond angles, and torsion angles as input, requiring the model to predict the same substructures generated by the DFT conformation, as well as the target objectives of the current dataset.

## 4 EXPERIMENTS

The experimental section aims to validate the effectiveness of our proposed model and methods in addressing existing challenges. We first demonstrate the performance and scalability of AGT on large-scale quantum chemistry datasets, PCQM4Mv2 (Hu et al., 2022) and OC20 (Chanussot et al., 2021). We then evaluate the transfer learning capabilities of the AGT model in both the conformer prediction and pre-training stages. We also conduct ablation studies on several key components of AGT and analyze different approaches to AGT's aggregated angle representation. Finally, quantitative analysis and visualization of conformer accuracy demonstrate that our proposed AGT model, compared to TGT, can distinguish chirality and more accurately predict bond angles and torsion angles, generating conformers that more closely resemble high-precision DFT conformers. The model is implemented using the PyTorch (Paszke et al., 2019) library. We perform mixed-precision training on 2 nodes, each equipped with 8 NVIDIA Tesla A100 GPUs (80GB RAM/GPU) and 16-core 2.6GHz Intel Xeon CPUs (320GB RAM per node).

### 4.1 LARGE-SCALE QUANTUM CHEMICAL PREDICTION

**PCQM4Mv2** PCQM4Mv2, part of the OGB-LSC graph property prediction challenge, contains over 3.7 million molecules. The dataset task is to predict the HOMO-LUMO gap. The performance

Table 2: Performance on OC20 IS2RE validation set.

| Model | Energy MAE (meV)↓ | | | | | EwT (%)↑ | | | | |
|---|---|---|---|---|---|---|---|---|---|---|
| | ID | OOD Ads. | OOD Cat. | OOD Both | AVG. | ID | OOD Ads. | OOD Cat. | OOD Both | AVG. |
| SchNet (Schütt et al., 2017) | 646.5 | 707.4 | 647.5 | 662.6 | 666.0 | 2.96 | 2.22 | 3.03 | 2.38 | 2.65 |
| DimeNet++ (Gasteiger et al., 2020) | 563.6 | 712.7 | 561.2 | 649.2 | 621.7 | 4.25 | 2.48 | 4.40 | 2.56 | 3.42 |
| GemNet-T (Gasteiger et al., 2021) | 556.1 | 734.2 | 565.9 | 696.4 | 638.2 | 4.51 | 2.24 | 4.37 | 2.38 | 3.38 |
| SphereNet (Liu et al., 2022b) | 563.2 | 668.2 | 559.0 | 619.0 | 602.4 | 4.56 | 2.70 | 4.59 | 2.70 | 3.64 |
| GNS (Godwin et al., b) | 540.0 | 650.0 | 550.0 | 590.0 | 582.5 | - | - | - | - | - |
| GNS+NN (Godwin et al., b) | 470.0 | 510.0 | 480.0 | 460.0 | 480.0 | - | - | - | - | - |
| Graphormer-3D (Shi et al., 2022) | 432.9 | 585.0 | 444.1 | 529.9 | 498.0 | - | - | - | - | - |
| EquiFormer (Liao & Smidt) | 422.2 | 542.0 | 423.1 | 475.4 | 465.7 | 7.23 | 3.77 | 7.13 | 4.10 | 5.56 |
| EquiFormer+NN (Liao & Smidt) | 415.6 | 497.6 | 416.5 | 434.4 | 441.0 | 7.47 | 4.64 | 7.19 | 4.84 | 6.04 |
| DRFormer (Wang et al., 2023) | 418.7 | 486.3 | 432.1 | 433.2 | 442.5 | 8.39 | 5.42 | 8.12 | 5.44 | 6.84 |
| Uni-Mol+ (Lu et al., 2023) | 379.5 | 452.6 | 401.1 | 402.1 | 408.8 | 11.1 | 6.71 | 9.90 | 6.68 | 8.61 |
| TGT (Hussain et al., 2024) | 381.3 | 445.4 | 391.7 | **393.6** | 403.0 | 11.1 | 6.87 | 10.47 | **6.80** | 8.82 |
| AGT | **377.2** | **441.3** | **384.6** | 394.9 | 399.5 | **11.2** | 6.95 | 11.26 | 6.79 | **8.99** |

Table 3: Results (MAE(↓)) on the QM9 dataset.

| Method | $\mu$ | $\alpha$ | $\epsilon_H$ | $\epsilon_L$ | $\Delta_\epsilon$ | ZPVE | $C_v$ |
|---|---|---|---|---|---|---|---|
| GraphMVP (Liu et al.) | 0.031 | 0.070 | 28.5 | 26.3 | 46.9 | 1.63 | 0.033 |
| GEM (Fang et al., 2022) | 0.034 | 0.081 | 33.8 | 27.7 | 52.1 | 1.73 | 0.035 |
| 3D Infomax (Stärk et al., 2022) | 0.034 | 0.075 | 29.8 | 25.7 | 48.8 | 1.67 | 0.033 |
| 3D-MGP (Jiao et al., 2023) | 0.020 | 0.057 | 21.3 | 18.2 | 37.1 | 1.38 | 0.026 |
| DimeNet++ (Gasteiger et al., 2020) | 0.030 | 0.044 | 24.6 | 19.5 | 32.6 | 1.21 | 0.023 |
| PaiNN (Schütt et al., 2021) | 0.012 | 0.045 | 27.6 | 20.4 | 45.7 | 1.28 | 0.024 |
| EGNN (Satorras et al., 2021) | 0.029 | 0.071 | 29.0 | 25.0 | 48.0 | 1.55 | 0.031 |
| SphereNet (Liu et al., 2022b) | 0.025 | 0.053 | 22.8 | 18.9 | 31.1 | 1.12 | 0.024 |
| EQGAT (Le et al., 2022) | 0.011 | 0.053 | 20.0 | 16.0 | 32.0 | 2.00 | 0.024 |
| ComENet (Wang et al., 2022) | 0.025 | 0.045 | 23.1 | 19.8 | 32.4 | 1.20 | 0.024 |
| LEFTNet (Du et al., 2024) | 0.011 | 0.039 | 23 | 18 | 39 | 1.19 | 0.022 |
| SaVeNet (Aykent & Xia, 2024) | **0.0085** | **0.035** | 16.6 | 15.1 | 22.7 | **1.10** | 0.021 |
| SE(3)-T (Fuchs et al., 2020) | 0.051 | 0.142 | 35.0 | 33.0 | 53.0 | - | 0.052 |
| TorchMD-Net (Thölke & De Fabritiis, 2022) | 0.011 | 0.059 | 20.3 | 17.5 | 36.1 | 1.84 | 0.026 |
| Equiformer (Liao & Smidt) | 0.011 | 0.046 | 15.0 | 14.0 | 30.0 | 1.26 | 0.023 |
| EquiformerV2 (Liao et al., 2024) | 0.010 | 0.050 | 14 | 13 | 29 | 1.47 | 0.023 |
| EquiformerV2+NN (Liao et al., 2024) | 0.009 | 0.039 | 12.2 | 11.4 | 24.2 | 1.21 | **0.020** |
| Transformer-M (Luo et al., 2022) | 0.037 | 0.041 | 17.5 | 16.2 | 27.4 | 1.18 | 0.022 |
| Geoformer (Wang et al., 2024a) | 0.010 | 0.040 | 18.4 | 15.4 | 33.8 | 1.28 | 0.022 |
| TGT (Hussain et al., 2024) | 0.025 | 0.040 | 9.9 | 9.7 | 17.4 | 1.18 | **0.020** |
| AGT | 0.019 | 0.037 | **8.8** | **9.1** | **16.4** | 1.14 | **0.020** |

of the distance predictor is tuned on a random 5% subset of the training data, which we refer to as validation-3d. Training the AGT model requires approximately 38 A100 GPU days, a 20% increase compared to the 32 A100 GPU days for TGT training, but still less than the 40 A100 GPU days required for UniMol+. Experimental results, expressed as Mean Absolute Error (MAE) in meV, are presented in Table 1. We observe that the 24-layer AGT model achieves the best performance on the PCQM4Mv2 dataset, surpassing the previous state-of-the-art TGT model by 0.9 meV. Notably, local chirality primarily affects molecular spatial configuration rather than electronic structure, so the prediction target (HOMO-LUMO gap) in PCQM4Mv2 has limited correlation with molecular local chirality. The enhanced local chirality expression capability of the AGT model compared to the TGT model provides minimal assistance in this task. Nevertheless, AGT still outperforms TGT on this dataset through more accurate prediction of torsion angles. The 24-layer AGT currently ranks first on the PCQM4Mv2 leaderboard, surpassing all baseline models, demonstrating the effectiveness of our proposed model. The 12-layer AGT model also exhibits strong performance, second only to the 24-layer TGT and AGT. The gap between the 12-layer and 24-layer AGT suggests that effectively encoding higher-order substructures on graphs requires deeper model architectures and larger model capacities.

**Open Catalyst 2020 IS2RE** The Open Catalyst 2020 Challenge aims to predict the adsorption energy of molecules on catalyst surfaces. We conduct experiments on the IS2RE (Initial Structure to Relaxed Energy) task. The IS2RE dataset provides initial DFT structures of crystals and adsorbates, which interact to reach a relaxed structure when measuring relaxed energy of the system. Following

Table 4: LIT-PCBA results.

| Model | Avg. Test ROC-AUC↑ (%) |
|---|---|
| NaiveBayes (Webb et al., 2010) | 73.0 |
| SVM (Hearst et al., 1998) | 73.4 |
| RandomForest (Breiman, 2001) | 62.0 |
| XGBoost (Chen & Guestrin, 2016) | 72.6 |
| GCN (Kipf & Welling, 2016) | 72.3 |
| GAT (Velickovic et al., 2018) | 75.2 |
| FP-GNN (Cai et al., 2022) | 75.9 |
| EGT (Hussain et al., 2022) | 78.9 |
| GEM (Fang et al., 2022) | 78.4 |
| GEM-2 (Liu et al., 2022a) | 81.5 |
| EGT+RDKit (Hussain et al., 2024) | 81.2 |
| TGT (Hussain et al., 2024) | 81.5 |
| AGT | **81.8** |

Table 5: Result on MOLPCBA and MOLHIV.

| Model | MOLPCBA | MOLHIV |
|---|---|---|
| | Test AP(%)↑ | Test ROC-AUC(%)↑ |
| DeeperGCN-VN (Li et al., 2020) | 28.42 $_{(0.43)}$ | 79.42$_{(1.20)}$ |
| PNA (Corso et al., 2020) | 28.38 $_{(0.35)}$ | 79.05$_{(1.32)}$ |
| DGN (Beaini et al., 2021) | 28.85 $_{(0.30)}$ | 79.70$_{(0.97)}$ |
| GINE-VN (Brossard et al., 2020) | 29.17 $_{(0.15)}$ | 77.10$_{(1.50)}$ |
| PHC-GNN (Le et al., 2021) | 29.47 $_{(0.26)}$ | 79.34$_{(1.16)}$ |
| GIN-VN$_{pretrain}$ (Gilmer et al., 2017) | 29.02 $_{(0.17)}$ | 77.07$_{(1.19)}$ |
| Graphormer (Ying et al., 2021) | 31.40 $_{(0.34)}$ | 80.51$_{(0.53)}$ |
| EGT (Hussain et al., 2022) | 29.61 $_{(0.24)}$ | 80.60$_{(0.65)}$ |
| TGT (Hussain et al., 2024) | 31.67 $_{(0.31)}$ | 80.71$_{(0.48)}$ |
| AGT | **31.79** $_{(0.26)}$ | **81.06**$_{(0.39)}$ |

Table 6: Distance and angle prediction performance of different edge-angle interaction mechanisms and training times on PCQM4Mv2.

| | No Angle Attention | No Edge-Angle Interaction | Total Edge-Angle Interaction | Topological Edge-Angle Interaction | Axial Edge-Angle Interaction | Geometric Edge-Angle Interaction |
|---|---|---|---|---|---|---|
| Dist. Cross-Ent.(↓) | 1.204 | 1.202 | 1.179 | 1.171 | 1.164 | **1.151** |
| Angle Cross-Ent.(↓) | - | 1.375 | 1.307 | 1.283 | 1.310 | **1.268** |
| Time/Epoch(↓) | **1.00** | 1.17 | 1.43 | 1.21 | 1.36 | 1.24 |

TGT's experimental configuration, we crop/sample based on the distance to adsorbate atoms, limiting the number of atoms to a maximum of 64. Training the model requires approximately 38 A100 GPU days. Due to additional angle constraint optimization, it requires slightly more training time compared to TGT, but still significantly less than the 112 GPU days used by UniMol+. Results for the IS2RE task are presented in Table 2, expressed as MAE (in meV) and Energy within Threshold (EwT) at 20 meV. The table shows that AGT achieves state-of-the-art (SOTA) performance on most subsets of the IS2RE evaluation dataset without significantly increasing computational resources. Specifically, it outperforms current methods on the ID (In Domain) and OOD (Out of Domain) Adsorbates and Catalyst subsets, while performing comparably to TGT on the OOD Both subset. Overall, our AGT model demonstrates superior average performance compared to the SOTA TGT model, securing its position as the best-performing direct method on the OC20 IS2RE task.

## 4.2 TRANSFER LEARNING

Our model learns two distinct forms of knowledge in two stages during large-scale training on the PCQM4Mv2 dataset. In the conformer prediction stage, the conformer predictor learns geometric information by predicting high-precision conformations. In the pre-training stage, the task predictor learns the quantum chemical properties of molecules by predicting the HOMO-LUMO gap. Therefore, in this section, we validate the transfer learning effectiveness of these two types of knowledge learned by AGT.

**Finetuning on QM9** We fine-tuned the task predictor of PCQM4Mv2 in the QM9 data set. This dataset allows the use of precise 3D conformational information during inference, so the task predictor only needs to train. We report the fine-tuning performance on a subset of 7 tasks out of 12 in QM9. See Appendix 14 for full results. As shown in Table 3, AGT achieves state-of-the-art results and, like TGT, significantly outperforms other models in predicting HOMO($\epsilon_H$), LUMO($\epsilon_L$), and HOMO-LUMO gap($\Delta_\epsilon$) - three tasks directly related to the pre-training task. Notably, AGT surpasses TGT in 6 of these tasks and performs comparably in the remaining one. This demonstrates that AGT's utilization of geometric information more effectively facilitates positive knowledge transfer to these tasks.

**Molecular Property Prediction** For the MOLPCBA (Hu et al., 2020) and MOLHIV molecular property prediction and LIT-PCBA (Tran-Nguyen et al., 2020) drug discovery benchmarks, we provide predictions of interatomic distances, bond angles, and torsion angles. These datasets lack ground truth 3D information. Therefore, we employ AGT's pre-trained conformer predictor as a frozen feature extractor. Results for MOLPCBA and MOLHIV are presented in Table 5. For MOLPCBA, the test mean Average Precision (%) is reported for a multi-task setting predicting 128 different binary molecular properties. For MOLHIV, the test ROC-AUC (%) is reported, indicating

Table 7: Ablation Study on PCQM4Mv2.

| AGT Att. Module | Directed Cycle Loss | Hiera. Virtual Node | Mode Distribution $(p_{Distance}, p_{Angle})$ | Val. MAE↓ (meV) |
|---|---|---|---|---|
| - | - | - | - | 73.6 |
| ✓ | - | - | - | 71.3 |
| ✓ | ✓ | - | - | 70.8 |
| ✓ | ✓ | ✓ | 1:1 | 70.3 |
| ✓ | ✓ | ✓ | 1:2 | 70.7 |
| ✓ | ✓ | ✓ | 2:1 | 69.8 |
| ✓ | ✓ | ✓ | 4:1 | 69.1 |
| ✓ | ✓ | ✓ | 8:1 | 70.4 |

the model's ability to predict whether a molecule inhibits HIV virus replication or not. As shown in the table, using the conformer predictor from the pre-trained AGT model yields the best results, surpassing TGT and significantly outperforming other pre-trained models. For the LIT-PCBA dataset, we report the average ROC-AUC (%) across 7 separate tasks predicting protein interactions in Table 4. We observe that AGT surpasses other pre-trained models, achieving state-of-the-art results. These experiments indicate that our pre-trained AGT's conformer predictor can provide more valuable 3D information to the task predictor for downstream tasks compared to RDKit coordinates, even when trained on a different dataset.

**Ablation Study** Table 6 compares the impact of different interaction methods between substructures of various orders in the AGT module on interatomic distance prediction, angle prediction in conformations, and training time. We use cross-entropy loss on the PCQM4Mv2 validation-3D set as the metric for distances and angles. Total edge-angle interaction refers to information exchange between bond angle and torsion angle structures with all pairwise embeddings. Axial edge-angle interaction involves interaction with pairwise embeddings that share common atoms with the endpoints of angle structures. Topological edge-angle interaction selects pairwise embeddings corresponding to edges in the 2D molecular topology graph for interaction. Geometric edge-angle interaction communicates with pairwise embeddings corresponding to the edges of the triangle containing the bond angle and the edges of the tetrahedron containing the torsion angle. We observe that geometric edge-angle interaction performs best in both distance and angle predictions, with a relatively low time cost among all variants. Notably, total edge-angle interaction is the most time-consuming but performs poorly, while axial edge-angle interaction, which reduces interaction objects, improves prediction performance. This suggests that interaction between higher-order and lower-order substructures requires finding the most relevant representations.

Table 7 presents an ablation study on our three main optimization designs and the ratio of distance to angle loss in the objective function. The results are from a 12-layer AGT model on PCQM4Mv2. We observe that the addition of the AGT module brings significant improvements. When learning angle information, the Directed Cycle Angle Loss helps reduce optimization difficulty for the model. The hierarchical virtual nodes in the task predictor serve as intermediate representations, aggregating and transmitting features from different levels of graph structures, providing a richer information basis for the final prediction task. Lastly, we experimented with different ratios of distance loss to angle loss and found that the model performs best when the ratio is 1:4.

## 5 CONCLUSION

In this work, we introduce the AGT architecture, which directly models higher-order substructures such as bond angles and torsion angles in molecular graphs, significantly enhancing the expressiveness and accuracy of molecular geometry modeling. We propose efficient interaction mechanisms between substructures of different orders and an angle objective function optimized for local chirality. Furthermore, we employ hierarchical virtual nodes in the task predictor, mitigating information compression of critical structures and neglect of geometric structures in property prediction. Through extensive experiments, we demonstrate state-of-the-art prediction accuracy on quantum chemistry datasets, as well as the transfer learning capabilities of both the conformation predictor and task predictor. In future work, we plan to explore inequality relationships and dynamic change representations of higher-order substructures in spatial stereochemistry, enabling more effective and rational geometric constraints for structural predictions.

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

## A  DENSITY FUNCTIONAL THEORY FOR MOLECULAR CONFORMATION PREDICTION

Density Functional Theory (DFT) (Kohn et al., 1996; Orio et al., 2009) is a first-principles computational method based on quantum mechanics that plays a crucial role in molecular conformation generation and property prediction. DFT describes many-electron systems through electron density rather than wave functions, significantly reducing computational complexity. Its theoretical foundation rests on the Hohenberg-Kohn theorem, which proves that all properties of a system's ground state can be uniquely determined by the electron density. In practical applications, the complex many-electron problem is transformed into more tractable single-electron problems through the Kohn-Sham equations. In molecular conformation generation, DFT can obtain precise three-dimensional conformations of molecules by solving electronic structure equations. This process includes optimizing molecular geometry, calculating bond lengths, bond angles, and dihedral angles, determining the lowest energy conformation, and predicting electron distribution within molecules. The molecular conformations generated by DFT possess high accuracy and are often used as benchmarks for the evaluation of other conformation generation methods. This high precision stems from its rigorous quantum mechanical theoretical foundation, which can accurately describe electronic effects, chemical bonding properties, and intramolecular interactions in molecules. However, DFT calculations also have limitations, such as high computational cost and difficulty in handling large molecular systems. In modern molecular design, DFT often complements machine learning methods (Schütt et al., 2017; Axelrod & Gomez-Bombarelli, 2022; Smith et al., 2020). Machine learning models can quickly predict molecular properties and initial conformations, while DFT is used to generate high-precision reference conformations and validate results. This combination leverages the advantages of both methods: the efficiency of machine learning and the high accuracy of DFT. With improvements in computational power and algorithms, DFT's applications in molecular science research will continue to expand, providing crucial support for drug design, materials development, and other fields.

## B  QUANTITATIVE ANALYSIS OF CHIRALITY PREDICTION

The conformation predictor outputs binned distances and angles under local chirality constraints, providing essential structural information for downstream task predictors. To quantitatively evaluate AGT's improvement over TGT in handling local chirality, we conducted a systematic evaluation on the PCQM4M training set, which contains 3,803,453 molecules, including 1,772,922 molecules with chiral centers (46.61%). The evaluation methodology compares model-predicted 3D conformers with high-precision DFT-calculated conformers, using angular deviations around chiral centers as the assessment criterion, with a deviation threshold of $\pi/6$. Experimental results demonstrate AGT's superiority over the baseline TGT model across three key metrics in Table 8. In terms of bond angle MAE, AGT achieves 0.209 rad, a 15.0% reduction compared to TGT's 0.246 rad. For torsion angle MAE, AGT reaches 0.334 rad, significantly lower than TGT's 0.597 rad by 44.1%. Regarding chirality prediction accuracy, AGT attains 74.7%, substantially outperforming TGT's 32.5% with a 130% improvement. These quantitative results strongly validate AGT's excellence in modeling chiral structures, particularly in complex torsion angle prediction and overall chirality determination tasks. The substantial improvements across all metrics demonstrate the effectiveness of AGT's direct angle modeling approach in capturing local molecular geometry.

Table 8: Comparison of AGT and TGT performance on chirality prediction

| Model | Bond Angles MAE (rad) | Torsion Angles MAE (rad) | Chirality Pred (%) |
|-------|-----------------------|--------------------------|--------------------|
| TGT   | 0.246                 | 0.597                    | 32.5               |
| AGT   | **0.209**             | **0.334**                | **74.7**           |

## C  THE ACCURACY OF CONFORMATION PREDICTOR IN ANGLES AND DISTANCES

To demonstrate the accuracy of AGT in geometric conformation prediction, we convert distances and angles to continuous unbounded values. Following the strategy employed in TGT (Hussain et al., 2024), we train two small refinement networks for distances and angles respectively. These networks accept clipped and binned values as input and output continuous, unbounded values. We train these networks using MAE loss and employ random inference to obtain the median of the output distances. We compare the accuracy of individual pairwise distances and angles on the validation-3D split of the PCQM4Mv2 dataset (i.e., data unseen during training), based on MAE, RMSE (Root Mean Square Error), and percentage errors within different thresholds as shown in Table 9 and Table 10. Our findings indicate that in terms of distances, our AGT predictor outperforms TGT across all metrics. Regarding angles, AGT significantly surpasses both RDKit and TGT in bond angle prediction and substantially leads in torsion angle prediction. This suggests that through angle constraints, AGT's conformation predictor can more accurately predict the underlying structure of molecules compared to the distance predictor in TGT.

Table 9: Accuracy of pairwise distances in terms of MAE↓, RMSE↓ and percent error within a threshold (EwT↑).

| Model | MAE (Å) | RMSE (Å) | EwT-0.2Å(%) | EwT-0.1Å(%) | EwT-0.05Å(%) | EwT-0.01Å(%) |
|---|---|---|---|---|---|---|
| RDKit | 0.248 | 0.541 | 73.33 | 66.65 | 56.90 | 26.79 |
| TGT + Refiner | 0.152 | 0.378 | 80.53 | 75.68 | 70.80 | 54.54 |
| AGT + Refiner | **0.131** | **0.327** | **86.74** | **78.51** | **74.09** | **57.17** |

Table 10: Accuracy of bond angles and torsion angles in terms of MAE↓, RMSE↓ and percent error within a threshold (EwT↑).

| **Model** | Bond Angles | | | Torsion Angles | | |
|---|---|---|---|---|---|---|
| | MAE (rad) | RMSE (rad) | EwT-$\pi/16$ rad (%) | MAE (rad) | RMSE (rad) | EwT-$\pi/16$ rad (%) |
| RDKit | 0.239 | 0.575 | 71.43 | 0.694 | 1.145 | 33.62 |
| TGT + Refiner | 0.225 | 0.431 | 76.26 | 0.563 | 0.713 | 41.89 |
| AGT + Refiner | **0.191** | **0.380** | **82.31** | **0.329** | **0.490** | **60.51** |

## D  EFFICIENCY ANALYSIS OF AGT VERSUS BASELINE MODELS

### D.1  PCQM4Mv2

Table 11 presents a comprehensive comparison of AGT against state-of-the-art molecular pre-training methods, Unimol+ and TGT, across different model scales, showing parameter counts, computational complexity, experimental performance on the PCQM4Mv2 dataset, and training/inference times. Based on experimental results, we comprehensively analyze AGT's method from both efficiency and effectiveness perspectives. Regarding computational complexity, where N represents the number of atoms, AGT requires $O(N^3)$ complexity for standard atom and pair embedding interactions, plus additional interactions between bond angles and torsion angles. In typical molecules, the number of bond angles ranges from 1.5N to 2N, and torsion angles from N to 2N, resulting in an additional computational complexity of $O(N^2)$, yielding an overall complexity of $O(N^3) + O(N^2)$.

On the large-scale PCQM4Mv2 dataset, AGT demonstrates an excellent balance between performance and computational efficiency. We systematically analyzed the trade-off between model scale and performance. Results show that 6-layer AGT (68M parameters) achieves an MAE of 69.4 meV,

comparable to 18-layer Unimol+ (77M parameters) at 69.3 meV, while significantly reducing training time (approximately 14 days versus 40 days using A100 GPU). As model layers increase, 24-layer AGT (241M parameters) reduces MAE to 66.2 meV, significantly outperforming 24-layer TGT (203M parameters, 67.1 meV MAE). Notably, although AGT's theoretical complexity is slightly higher than baseline models, 12-layer AGT (127M parameters) maintains competitive performance (69.1 meV MAE) while reducing training time from 38 to 20 GPU days, with corresponding inference time reduction. These results indicate that AGT architecture is competitive even at smaller scales and can better leverage its structural modeling advantages as parameter count increases.

### D.2 OPEN CATALYST 2020 IS2RE

Table 12 presents a comprehensive evaluation of AGT against both pre-trained and non-pre-trained methods on the OC20 dataset, focusing on computational efficiency and model performance. Based on experimental results, we analyze AGT's capabilities from multiple perspectives. Regarding computational efficiency, AGT demonstrates competitive inference and fine-tuning times compared to non-pre-training methods. Specifically, AGT's fine-tuning duration (240 minutes) aligns well with established models such as DimeNet++ (230 minutes), GemNet-T (200 minutes), and SphereNet (290 minutes). While ComENet exhibits faster training speed (20 minutes), AGT achieves substantially superior performance metrics, with energy MAE of 399.5 meV versus 588.8 meV and FwT of 8.99% versus 3.56%, validating the effectiveness of our pre-training strategy. In comparison with other pre-trained methods, AGT shows remarkable efficiency improvements while maintaining performance advantages. Compared to TGT, despite incorporating additional angular information and direct angle modeling mechanisms, AGT maintains similar training efficiency (approximately 34 days versus 32 days using A100 GPU) while achieving superior performance. Notably, compared to Uni-Mol+, AGT achieves better performance metrics while significantly reducing pre-training time (34 days versus 112 days using A100 GPU), demonstrating an optimal balance between computational efficiency and model effectiveness.

Table 11: Comparison of performance and efficiency metrics on PCQM4Mv2

| Model | # param. | Complexity | # layers | MAE (meV) | Training Time | Inference Time |
|---|---|---|---|---|---|---|
| Unimol+ | 27.7M | $O(N^3)$ | 6 | 71.4 | - | - |
| Unimol+ | 52.4M | $O(N^3)$ | 12 | 69.6 | - | - |
| Unimol+ | 77M | $O(N^3)$ | 18 | 69.3 | ~40 A100 GPU day | ~56 V100 GPU min |
| TGT | 116M | $O(N^3)$ | 12 | 70.9 | - | - |
| TGT | 203M | $O(N^3)$ | 24 | 67.1 | ~32 A100 GPU day | ~40 A100 GPU min |
| AGT | 68M | $O(N^3) + O(N^2)$ | 6 | 69.4 | ~14 A100 GPU day | ~19 A100 GPU min |
| AGT | 127M | $O(N^3) + O(N^2)$ | 12 | 69.1 | ~20 A100 GPU day | ~31 A100 GPU min |
| AGT | 241M | $O(N^3) + O(N^2)$ | 24 | **66.2** | ~38 A100 GPU day | ~40 A100 GPU min |

Table 12: Comparison of performance and efficiency metrics on OC20

| Model | Pretraining Time | Train Time | Inference Time | Avg. Energy MAE (meV) ↓ | Avg. FwT (%) ↑ |
|---|---|---|---|---|---|
| CGCNN | - | 18min | 1min | 658.5 | 2.82 |
| SchNet | - | 10min | 1min | 666.0 | 2.65 |
| DimeNet++ | - | 230min | 4min | 621.7 | 3.42 |
| GemNet-T | - | 200min | 4min | 638.2 | 3.38 |
| SphereNet | - | 290min | 5min | 602.3 | 3.64 |
| ComENet | - | 20min | 1min | 588.8 | 3.56 |
| Unimol+ | 112 A100 GPU days | - | - | 408.8 | 8.61 |
| TGT | 32 A100 GPU days | - | - | 403.0 | 8.82 |
| AGT | 34 A100 GPU days | 240min | 7min | **399.5** | **8.99** |

## E EXPERIMENTAL DETAILS

The hyperparameters used for each dataset are presented in Table E. For PCQM4Mv2 and OC20 we list the hyperparameters for both the conformation and the task predictor models and both training

Table 13: Hyperparameters for each dataset.

| Hyperparameters | PCQM4Mv2 | | OC20 | | QM9 | MOLPCBA | LIT-PCBA | MOLHIV |
|---|---|---|---|---|---|---|---|---|
| | Conf. Pred. | Task Pred. | Conf. Pred. | Task Pred. | Task Pred. | Task Pred. | Task Pred. | Task Pred. |
| # Layers | 24 | 24 | 24 | 14 | 24 | 12 | 8 | 12 |
| Node Embed. Dim | 768 | 768 | 768 | 768 | 768 | 768 | 1024 | 768 |
| Edge Embed. Dim | 256 | 256 | 256 | 512 | 256 | 32 | 256 | 32 |
| Angle Embed. Dim | 128 | 128 | 128 | 256 | 128 | 32 | 128 | 32 |
| # Attn. Heads | 64 | 64 | 64 | 64 | 64 | 32 | 64 | 32 |
| # Triplet Heads | 16 | 16 | 16 | 16 | 16 | 4 | 0 | 4 |
| Node FFN Dim. | 768 | 768 | 1536 | 768 | 768 | 768 | 2048 | 768 |
| Edge FFN Dim. | 256 | 256 | 512 | 512 | 256 | 32 | 512 | 32 |
| Angle FFN Dim. | 128 | 128 | 256 | 256 | 128 | 32 | 256 | 32 |
| Max. Hops Enc. | 32 | 32 | - | - | 32 | 32 | 32 | 32 |
| Activation | GELU | GELU | GELU | GELU | GELU | GELU | GELU | GELU |
| Input Dist. Enc. | RBF | RBF | Fourier | Fourier | RBF | RBF | RBF | RBF |
| Source Dropout | 0.3 | 0.3 | 0.3 | 0.3 | 0.3 | 0.3 | 0.3 | 0.3 |
| Triplet Dropout | 0.0 | 0.0 | 0.1 | 0.0 | 0.0 | 0.1 | 0.0 | 0.0 |
| Path Dropout | 0.2 | 0.2 | 0.2 | 0.1 | 0.2 | 0.1 | 0.1 | 0.1 |
| Node Activ. Dropout | 0.1 | 0.1 | 0.1 | 0.1 | 0.1 | 0.1 | 0.1 | 0.1 |
| Edge Activ. Dropout | 0.1 | 0.1 | 0.1 | 0.1 | 0.1 | 0.1 | 0.1 | 0.1 |
| Angle Activ. Dropout | 0.1 | 0.1 | 0.1 | 0.1 | 0.1 | 0.1 | 0.1 | 0.1 |
| Input 3D Noise | - | 0.2 | - | 0.6 | 0.0 | - | - | - |
| Input Noise Smooth. | - | 1.0 | - | 1.0 | 0.0 | - | - | - |
| Optimizer | Adam | Adam | Adam | Adam | Adam | Adam | Adam | Adam |
| Batch Size | 1024 | 2048 | 256 | 256 | - | 256 | 1024 | 256 |
| Max. LR | 0.001 | 0.0015 | 0.001 | 0.001 | - | $4 \times 10^{-4}$ | $5 \times 10^{-4}$ | $3 \times 10^{-4}$ |
| Min. LR | $10^{-6}$ | $10^{-6}$ | 0.001 | $10^{-6}$ | - | $10^{-8}$ | $5 \times 10^{-7}$ | $10^{-8}$ |
| Warmup Steps | 30000 | 20000 | 8000 | 16000 | - | 5000 | 600 | 5000 |
| Total Training Steps | 60000 | 350000 | 30000 | 100000 | - | 30000 | 1200 | 30000 |
| Grad. Clip. Norm | 5.0 | 5.0 | 5.0 | 5.0 | 5.0 | 5.0 | 2.0 | 5.0 |
| Conf. Loss Weight | - | 0.1 | - | 3.0 | 0.0 | 0.05 | 0.1 | 0.05 |
| # Angle Bins | 256 | 512 | 256 | 512 | - | 512 | 512 | 512 |
| # Dist. Bins | 256 | 512 | 256 | 512 | - | 512 | 512 | 512 |
| Dist. Bins Range | 8 | 8 | 16 | 16 | - | 8 | 8 | 8 |
| FT Batch Size | - | 2048 | - | 1024 | 2048 | - | - | - |
| FT Warmup Steps | - | 3000 | - | 0 | 3000 | - | - | - |
| FT Total Steps | - | 50000 | - | 12000 | 150000 | - | - | - |
| FT Max. LR | - | $2 \times 10^{-4}$ | - | $10^{-5}$ | $2 \times 10^{-4}$ | - | - | - |
| FT Min. LR | - | $10^{-6}$ | - | $10^{-5}$ | $10^{-6}$ | - | - | - |
| FT Conf. Loss Weight | - | 0.1 | - | 2.0 | 0.1 | - | - | - |

and finetuning. For QM9, we only list the hyperparameters for finetuning. For MOLPCBA, LIT-PCBA, and MOLHIV we only show the hyperparameters for training from scratch. The missing hyperparameters do not apply to the corresponding dataset or model. For QM9 no secondary distance and angle denoising objective is used. For LIT-PCBA, 0 triplet interaction heads indicate that an EGT is used without any triplet interaction module.

To provide the conformation predictor with initial 3D information, we utilize RDKit (Landrum, 2013) to extract 3D coordinates and apply MM Force Field Optimization (Halgren, 1996). Due to the absence of Ground Truth 3D coordinates in the the PCQM4Mv2 validation set, we randomly divide the training set into train-3D and validation-3D splits, with the latter containing 5% of the training data. Hyperparameters of the conformation predictor are fine-tuned by monitoring the average cross-entropy loss of binned distance and angle prediction on the validation-3D split, which is found to be a good indicator of downstream performance. The input noise level is adjusted by evaluating the finetuned performance on the validation set. We get the best results by using an average of 50 sample predictions during stochastic inference. Other training configurations not mentioned are based on TGT (Hussain et al., 2024).

# F    ADDITIONAL RESULTS AND ANALYSES

## F.1    QM9

In this appendix, we present the comprehensive evaluation results on the QM9 dataset across all 12 prediction tasks (see Table 14). The detailed performance analysis shows that AGT demonstrates strong predictive capabilities across various molecular properties. Particularly noteworthy are the

Table 14: Results (MAE($\downarrow$)) on the QM9 dataset.

| Method | $\mu$ | $\alpha$ | $\epsilon_H$ | $\epsilon_L$ | $\Delta\epsilon$ | ZPVE | $C_v$ | $U_0$ | $U$ | $H$ | $G$ | $R^2$ |
|---|---|---|---|---|---|---|---|---|---|---|---|---|
| GraphMVP (Liu et al.) | 0.031 | 0.070 | 28.5 | 26.3 | 46.9 | 1.63 | 0.033 | - | - | - | - | - |
| GEM (Fang et al., 2022) | 0.034 | 0.081 | 33.8 | 27.7 | 52.1 | 1.73 | 0.035 | - | - | - | - | - |
| 3D Infomax (Stärk et al., 2022) | 0.034 | 0.075 | 29.8 | 25.7 | 48.8 | 1.67 | 0.033 | - | - | - | - | - |
| 3D-MGP (Jiao et al., 2023) | 0.020 | 0.057 | 21.3 | 18.2 | 37.1 | 1.38 | 0.026 | - | - | - | - | - |
| Schnet (Schütt et al., 2017) | 0.033 | 0.235 | 41.0 | 34.0 | 63.0 | 1.7 | 0.033 | 14 | 19 | 14 | 14 | 73 |
| PhysNet (Unke & Meuwly, 2019) | 0.053 | 0.062 | 32.9 | 24.7 | 42.5 | 1.39 | 0.028 | 8.15 | 8.34 | 8.42 | 9.4 | 765 |
| Cormorant (Anderson et al., 2019) | 0.038 | 0.085 | 34.0 | 38.0 | 61.0 | 2.03 | 0.026 | 22 | 21 | 21 | 20 | 961 |
| DimeNet++ (Gasteiger et al., 2020) | 0.030 | 0.044 | 24.6 | 19.5 | 32.6 | 1.21 | 0.023 | 6.32 | 6.28 | 6.53 | 7.56 | 331 |
| PaiNN (Schütt et al., 2021) | _0.012_ | 0.045 | 27.6 | 20.4 | 45.7 | 1.28 | 0.024 | 5.85 | 5.83 | 5.98 | 7.35 | 66 |
| EGNN (Satorras et al., 2021) | 0.029 | 0.071 | 29.0 | 25.0 | 48.0 | 1.55 | 0.031 | 11 | 12 | 12 | 12 | 106 |
| NoisyNode (Godwin et al., a) | 0.025 | 0.052 | 20.4 | 18.6 | 28.6 | 1.16 | 0.025 | 7.30 | 7.57 | 7.43 | 8.30 | 700 |
| SphereNet (Liu et al., 2022b) | 0.025 | 0.053 | 22.8 | 18.9 | 31.1 | _1.12_ | 0.024 | 6.26 | 6.36 | 6.33 | 7.78 | 268 |
| ComENet (Wang et al., 2022) | 0.025 | 0.045 | 23.1 | 19.8 | 32.4 | 1.20 | 0.024 | 6.59 | 6.82 | 6.86 | 7.98 | 259 |
| SEGNN (Brandstetter et al., 2022) | 0.023 | 0.060 | 24.0 | 21.0 | 42.0 | 1.62 | 0.031 | 15 | 13 | 16 | 15 | 660 |
| EQGAT (Le et al., 2022) | 0.011 | 0.053 | 20.0 | 16.0 | 32.0 | 2.00 | 0.024 | 25 | 25 | 24 | 23 | 382 |
| LEFTNet (Du et al., 2024) | 0.011 | 0.039 | 23 | 18 | 39 | 1.19 | 0.022 | 5 | 5 | 5 | _6_ | 66 |
| SaVeNet (Aykent & Xia, 2024) | **0.0085** | **0.035** | 16.6 | 15.1 | 22.7 | **1.10** | 0.021 | 4.83 | 4.74 | 4.83 | 6.10 | 49 |
| SE(3)-T (Fuchs et al., 2020) | 0.051 | 0.142 | 35.0 | 33.0 | 53.0 | - | 0.052 | - | - | - | - | - |
| TorchMD-Net (Thölke & De Fabritiis, 2022) | 0.011 | 0.059 | 20.3 | 17.5 | 36.1 | 1.84 | 0.026 | 6.15 | 6.38 | 6.16 | 7.62 | _33_ |
| Equiformer (Liao & Smidt) | 0.011 | 0.046 | 15.0 | 14.0 | 30.0 | 1.26 | 0.023 | 6.59 | 6.74 | 6.63 | 7.63 | 251 |
| Transformer-M (Luo et al., 2022) | 0.037 | 0.041 | 17.5 | 16.2 | 27.4 | 1.18 | _0.022_ | 9.37 | 9.41 | 9.39 | 9.63 | 75 |
| TGT (Hussain et al., 2024) | 0.025 | 0.040 | _9.9_ | _9.7_ | _17.4_ | 1.18 | **0.020** | - | - | - | - | - |
| EquiformerV2 (Liao et al., 2024) | 0.010 | 0.050 | 14 | 13 | 29 | 1.47 | 0.023 | 6.17 | 6.49 | 6.22 | 7.57 | 186 |
| EquiformerV2+NN (Liao et al., 2024) | 0.009 | 0.039 | 12.2 | 11.4 | 24.2 | 1.21 | **0.020** | 4.34 | 4.28 | 4.24 | 5.34 | 182 |
| Geoformer (Wang et al., 2024a) | 0.010 | 0.040 | 18.4 | 15.4 | 33.8 | 1.28 | 0.022 | _4.43_ | _4.41_ | _4.39_ | 6.13 | **27.5** |
| AGT | 0.019 | _0.037_ | **8.8** | **9.1** | **16.4** | 1.14 | **0.020** | 6.33 | 6.52 | 6.59 | 6.94 | 70 |

Table 15: LIT-PCBA results in terms of ROC-AUC$\uparrow$ (%).

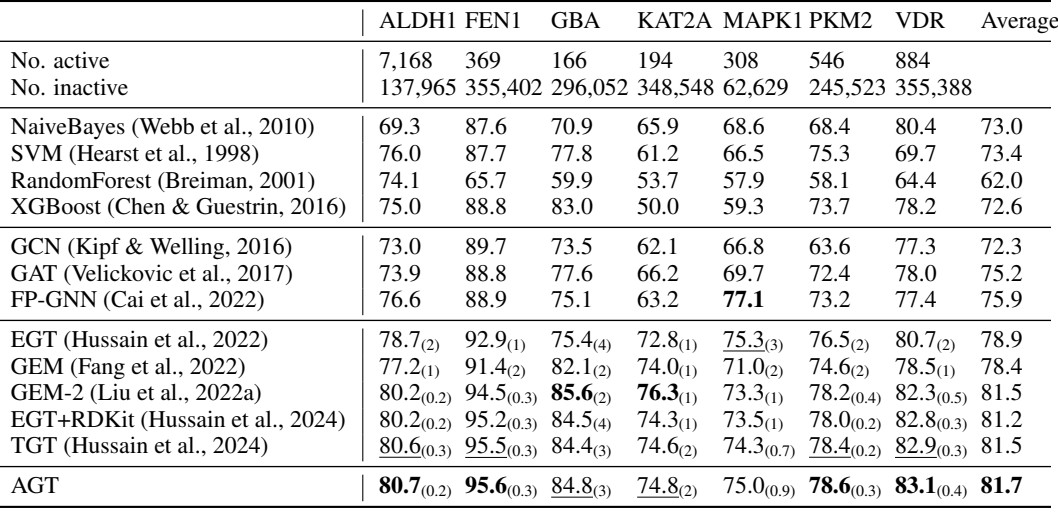

| | ALDH1 | FEN1 | GBA | KAT2A | MAPK1 | PKM2 | VDR | Average |
|---|---|---|---|---|---|---|---|---|
| No. active | 7,168 | 369 | 166 | 194 | 308 | 546 | 884 | |
| No. inactive | 137,965 | 355,402 | 296,052 | 348,548 | 62,629 | 245,523 | 355,388 | |
| NaiveBayes (Webb et al., 2010) | 69.3 | 87.6 | 70.9 | 65.9 | 68.6 | 68.4 | 80.4 | 73.0 |
| SVM (Hearst et al., 1998) | 76.0 | 87.7 | 77.8 | 61.2 | 66.5 | 75.3 | 69.7 | 73.4 |
| RandomForest (Breiman, 2001) | 74.1 | 65.7 | 59.9 | 53.7 | 57.9 | 58.1 | 64.4 | 62.0 |
| XGBoost (Chen & Guestrin, 2016) | 75.0 | 88.8 | 83.0 | 50.0 | 59.3 | 73.7 | 78.2 | 72.6 |
| GCN (Kipf & Welling, 2016) | 73.0 | 89.7 | 73.5 | 62.1 | 66.8 | 63.6 | 77.3 | 72.3 |
| GAT (Velickovic et al., 2017) | 73.9 | 88.8 | 77.6 | 66.2 | 69.7 | 72.4 | 78.0 | 75.2 |
| FP-GNN (Cai et al., 2022) | 76.6 | 88.9 | 75.1 | 63.2 | **77.1** | 73.2 | 77.4 | 75.9 |
| EGT (Hussain et al., 2022) | 78.7(2) | 92.9(1) | 75.4(4) | 72.8(1) | _75.3_(3) | 76.5(2) | 80.7(2) | 78.9 |
| GEM (Fang et al., 2022) | 77.2(1) | 91.4(2) | 82.1(2) | 74.0(1) | 71.0(2) | 74.6(2) | 78.5(1) | 78.4 |
| GEM-2 (Liu et al., 2022a) | 80.2(0.2) | 94.5(0.3) | **85.6**(2) | **76.3**(1) | 73.3(1) | 78.2(0.4) | 82.3(0.5) | 81.5 |
| EGT+RDKit (Hussain et al., 2024) | 80.2(0.2) | 95.2(0.3) | 84.5(4) | 74.3(1) | 73.5(1) | 78.0(0.2) | 82.8(0.3) | 81.2 |
| TGT (Hussain et al., 2024) | _80.6_(0.3) | _95.5_(0.3) | 84.4(3) | 74.6(2) | 74.3(0.7) | _78.4_(0.2) | _82.9_(0.3) | 81.5 |
| AGT | **80.7**(0.2) | **95.6**(0.3) | _84.8_(3) | _74.8_(2) | 75.0(0.9) | **78.6**(0.3) | **83.1**(0.4) | **81.7** |

results in energy-related metrics ($\varepsilon_H$: 8.8, $\varepsilon_L$: 9.1, $\Delta\varepsilon$: 16.4, achieving state-of-the-art performance) and physical properties ($C_v$: 0.020, matching TGT's performance). For optical and quantum properties such as $\alpha$ and ZPVE, AGT shows competitive performance near the top of the benchmark. The model also demonstrates robust performance in thermodynamic properties ($U_0$, $U$, $H$, $G$) and geometric features ($R^2$), surpassing previous pre-trained approaches including Transformer-M.

The inability to achieve comprehensive superiority across all metrics can be attributed to several factors. First, there may be a mismatch between pre-training objectives and specific task requirements. AGT's pre-training optimization primarily focuses on the holistic representation of molecular structures, which might not fully capture the detailed features required for certain physicochemical properties. For instance, the prediction of $\mu$ may require better characterization of atomic electronegativity differences. Second, the task-specific nature of certain property predictions may demand more specialized model architectures or loss function designs, which a general pre-trained model might struggle to accommodate. Notably, since the supervision signal during pre-training comes from the HOMO-LUMO gap in the PCQM4Mv2 dataset, the pre-trained model may exhibit

a natural bias towards metrics with similar distributions, such as $\varepsilon_H$, $\varepsilon_L$, and $\Delta\varepsilon$, potentially at the expense of other metrics. Finally, the optimization strategy involves inherent trade-offs: to maintain model generality, AGT's pre-training process may have made compromises in performance on certain specific tasks. This balance between generalization and task-specific optimization remains a fundamental challenge in molecular representation learning.

In Table 3 and its complete version Table 14, we categorize methods into three distinct groups. The first group comprises pre-trained GNN methods, including GraphMVP, GEM, 3D Infomax, and 3D-MGP. The second group consists of directly trained GNN methods, spanning from GraphMVP through SaVeNet. The third group encompasses Transformer-based methods from SE(3)-T through AGT, where we do not distinguish between pre-trained and non-pre-trained models due to their common large-scale training dataset.

### F.2   LIT-PCBA

We also show a breakdown of the LIT-PCBA results for the individual protein targets in Table 15. Notice that, AGT outperforms other models in ALDH1, FEN1, PKM2, and VDR. Despite the low number of positive samples, AGT ranked second among all models on GBA and KAT2A, surpassing TGT (Hussain et al., 2024) on all proteins target. we can analyze why AGT shows slightly lower performance on GBA, KAT2A, and MAPK1 compared to some other methods. For GBA, which has a relatively small dataset (166 active samples vs 296,052 inactive samples), the extreme class imbalance might affect AGT's performance, resulting in a score of 84.8% compared to GEM-2's 85.6%. Similarly, KAT2A and MAPK1 both have limited active samples (194 and 308 respectively) with significant class imbalance. The performance differences are relatively small - for KAT2A, AGT achieves 74.8% compared to GEM-2's 76.3%, and for MAPK1, AGT's 75.0% is close to the best performers. These marginal differences might be attributed to the specific structural characteristics of these proteins and the extreme class imbalance in their datasets, which could potentially benefit from more specialized handling of imbalanced data during model training.

In Tables 4 and 15, we present three groups of methods. The first group consists of traditional machine learning methods (NaiveBayes, SVM, RandomForest, XGBoost). The second group consists of directly trained GNNs (GCN, GAT, FP-GNN). The third group consists of pre-trained deep learning methods from EGT through TGT.

### F.3   PCQM4Mv2

In Table 1, we organize methods into three groups. The first group represents earlier methods, ranging from MLP-Fingerprint through GPS++. The second group includes current state-of-the-art methods (Transformer-M, Uni-Mol+, TGT) that incorporate 3D conformation perturbation and denoising prediction. The final group consists solely of our proposed AGT method.

### F.4   OC20

For Table 2, methods are divided into two main categories. The first group encompasses GNN methods from SchNet through GNS+NN, while the second group includes Transformer-based methods from Graphormer-3D through TGT.

### F.5   MolPCBA and MolHIV

In Table 5, we categorize methods into two groups. The first group includes directly trained GNN methods from DeeperGCN-VN through PHC-GNN, while the second group comprises pre-trained deep learning methods from GIN-VN through TGT.

## G   Analysis of AGT's Capabilities and Limitations

## G.1 Task-Specific Performance Analysis

Analysis of experimental results on the QM9 dataset reveals heterogeneous performance across different property prediction tasks. AGT demonstrates exceptional performance in energy-related metrics ($\varepsilon_H$: 8.8, $\varepsilon_L$: 9.1, $\Delta\varepsilon$: 16.4, all achieving state-of-the-art results) and certain physical properties ($C_v$: 0.020, matching TGT's performance). However, the variation in performance across different metrics can be attributed to several key factors. The pre-training optimization of AGT primarily emphasizes comprehensive molecular structure representation. This approach may not fully capture the specific features required for certain physicochemical properties, particularly evident in properties like $\mu$ that demand precise characterization of atomic electronegativity differences. Furthermore, certain property prediction tasks necessitate specialized architectural components or loss function designs that may not be optimally addressed by general pre-trained frameworks. Notably, the pre-training process on PCQM4Mv2 dataset, which focuses on HOMO-LUMO gap prediction, introduces a beneficial bias towards related downstream tasks. This explains AGT's superior performance on QM9's energy-level related metrics ($\varepsilon_H$, $\varepsilon_L$, $\Delta\varepsilon$), as these properties share similar underlying electronic structure characteristics with the HOMO-LUMO gap. The strong correlation between pre-training objectives and downstream task performance demonstrates both the effectiveness of transfer learning in capturing fundamental electronic properties and the potential task-specific limitations of the pre-training approach. Additionally, the maintenance of model generality during pre-training may necessitate performance compromises on specific tasks, reflecting the balance between general applicability and task-specific optimization.

## G.2 Scalability Analysis

Our comprehensive evaluation of AGT spans across datasets with significantly different molecular scales, including PCQM4Mv2 (mean: 15 atoms), downstream tasks MolHIV and MolPCBA (mean: 26 atoms), and larger-scale OC20 systems (approximately 80 atoms). Notably, AGT achieves state-of-the-art performance among pre-training methods across all these datasets, demonstrating robust scalability without performance degradation even on OC20 dataset where molecules contain substantially more atoms.

Theoretically, AGT's architecture poses no inherent limitations on molecular size processing. However, in practical applications, the scalability of molecular processing is primarily constrained by two fundamental factors. The primary limitation stems from GPU memory capacity, which defines the maximum processable molecular system size when handling 3D conformer data. This constraint is particularly relevant for large-scale molecular systems requiring extensive memory allocation. From an algorithmic perspective, the scalability challenges for large-scale molecular systems (e.g., proteins) primarily arise from the rapid growth of higher-order structures. This growth pattern introduces challenges: the computational complexity increases quadratically with the system size, and the attention mechanism tends to suffer from performance degradation due to averaging effects across an expanding interaction space. For such challenges, potential solutions could draw inspiration from recent advances in protein structure prediction, particularly the mechanisms employed in AlphaFold3 (Abramson et al., 2024). Local attention mechanisms or sliding window strategies could theoretically constrain the attention parameters of bond angles and torsion angles to focus only on the k-nearest neighboring structures of the same order. Such localized approaches would potentially optimize the computation of angular interactions while preserving the essential local geometric relationships that typically dominate molecular properties. These theoretical modifications could substantially reduce computational complexity while maintaining model effectiveness, as local structural correlations often carry the most relevant information for property prediction tasks.

These theoretical considerations suggest potential pathways for handling larger molecular systems through algorithmic optimizations and computational strategies. The current demonstrated scalability, combined with consistent performance across different molecular sizes, indicates promising applications across an expanded range of molecular systems, from small molecules to larger biochemical structures. Future exploration of these optimization strategies may enable the extension of AGT to more complex molecular systems while maintaining computational efficiency.

## H   ADDITIONAL DETAILS ABOUT RELATED WORKS

**Molecular Property Prediction** The remarkable performance of message-passing GNNs in predicting molecular properties has inspired a new generation of geometric and physics-aware neural networks, which maintain invariance or equivariance under 3D rotational and translational transformations. Early developments in this direction include SchNet (Schütt et al., 2017) and DimeNet (Gasteiger et al., 2020), which pioneered the use of distance-based convolution approaches. The field further evolved with the introduction of spherical methodologies, as exemplified by GemNet (Gasteiger et al., 2021), SphereNet (Liu et al., 2022b), ComENet (Wang et al., 2022), LEFTNet (Du et al., 2024), and SAVENet (Aykent & Xia, 2024), each incorporating various forms of angular information. This architectural evolution ultimately led to more sophisticated equivariant transformer designs, including Equiformer (Liao & Smidt), EquiformerV2 (Liao et al., 2024), TorchMD-Net (Thölke & De Fabritiis, 2022), and Geoformer (Wang et al., 2024a), which generalized the concept of equivariant aggregation. While these advances have significantly improved molecular representation learning, our work proposes a fundamentally different paradigm for modeling higher-order structures. Recent models like QuinNet (Wang et al., 2024c) and ViSNet (Wang et al., 2024b) have introduced four or five-atom interactions to enhance model expressiveness and accuracy. However, these methods primarily focus on local representations of atomic nodes and chemical bonds, capturing higher-order features implicitly through combinatorial operations between atom-level tokens. In contrast, our approach transforms higher-order graph structures into independent token representations, enabling direct learning and representation of structural patterns in molecules. This innovation is particularly crucial for model interpretability and effective utilization of expert prior knowledge. From an information propagation perspective, traditional methods require higher-order structural information (such as four-body and five-body interactions) to propagate gradually along the graph topology, creating significant information bottlenecks. As demonstrated in TGT research, even information exchange between adjacent embeddings faces restrictions. Our method addresses these limitations through direct structural token representation, not only avoiding these bottlenecks but also enabling efficient access and utilization of key higher-order information by all graph nodes, thereby providing a more effective framework for learning molecular structural information.

