# OpenReview forum: "Angle Graph Transformer: Capturing Higher-Order Structures for Accurate Molecular Geometry Learning"
_ICLR.cc/2025/Conference — Submitted to ICLR 2025_

### Official Review · Reviewer_4RQ8 · 2024-10-31

**Soundness:** 3
**Presentation:** 3
**Contribution:** 3
**Rating:** 8
**Confidence:** 3

**Summary:**

This work introduces the higher-order structures of molecules (third-order bond angles and fourth-order torsional angles) into the Graph Transformer network, treating them as independent tokens, and proposes the Angle Graph Transformer (AGT) network. Its main contributions are: a) The introduction of higher-order structures reduces the difficulty of predicting local chirality in molecules; b) A DCA loss function is introduced to predict more accurate bond angles and torsion angles from low-precision initial conformations (generated by rdkit), allowing predictions to align more closely with DFT-derived conformations; c) This framework achieves state-of-the-art results in molecular pre-training and fine-tuning across numerous downstream tasks. Overall, the methods proposed in this work are effective, and the experimental results are compelling.

**Strengths:**

1. The experimental results are very solid; both in pre-training and downstream fine-tuning, AGT achieved the best results across most metrics on multiple datasets.
2. To my knowledge, this work’s consideration and design for accurately modeling local chirality in molecules is relatively novel, as previous studies largely overlooked this issue. Additionally, the introduction of a new DCA loss function allows for more precise structural predictions, and ablation experiments have demonstrated its effectiveness. These two aspects can be heuristically applied to tasks such as molecular conformation generation and ground-state conformation prediction.
3. The innovative design of directly inputting angles as tokens into the network and using different virtual nodes (Hierarchical Virtual Node) for different substructure tokens is noteworthy.

**Weaknesses:**

1. I believe that the statement in the abstract, “AGT directly models directed bond angles and torsion angles, introducing higher-order structural representations to molecular graph learning for the first time,” as well as similar expressions in the main text, may contain overstated claims or lack clarity. To my knowledge, works like QuinNet[1] and ViSNet[2] also attempt to incorporate higher-order many-body interactions into networks to learn better force fields. Could the authors provide a clear explanation if my understanding is incorrect?
2. I have reservations about the claim in section 3.1 that “methods predicting only interatomic distances cannot uniquely determine bond and torsion angles in the counterclockwise direction, as both σ and 2π − σ can satisfy the same distance matrix in 3D space.” While I understand that there may be molecules where the interatomic distance matrix remains unchanged under chirality variations, I believe that in most cases, if the positions of atoms in a local structure change in 3D space, the interatomic distances between those atoms and those in the unchanged part of the structure will necessarily change. Therefore, it would be more convincing if the authors could provide a more visual representation of this point, a formal description and proof, or a detailed analysis of the influence of local chirality within a specific dataset.
3. On the PCQM4MV2 dataset, the performance improvement of AGT-127M compared to Uni-Mol+ -77M is only 0.2 (69.1 vs. 69.3), which I believe is not a favorable trade-off. However, both AGT-241M and AGT-127M show a significant performance gap when compared to TGT-203M. As the authors mentioned, the introduction of higher-order structural information may require larger model capacity, but this may not be user-friendly for the community.

[1] Wang, Zun, et al. "Efficiently incorporating quintuple interactions into geometric deep learning force fields." *Advances in Neural Information Processing Systems* 36 (2024).

[2] Wang, Yusong, et al. "Enhancing geometric representations for molecules with equivariant vector-scalar interactive message passing." Nature Communications 15.1 (2024): 313.

**Questions:**

as Weaknesses.1 & Weaknesses.2

---

> ### Author Response · Authors · 2024-11-20
> **Answer 1 for Weaknesses and Questions**
>
> Thank you for recognizing our work. We will address your questions below.
>
> **W1:** Thank you for raising questions about the accuracy of our paper's contributions. We believe our core innovation lies in proposing a fundamentally different paradigm for direct modeling of higher-order structures. While models like QuinNet and ViSNet indeed introduced four or even five-atom interactions to enhance model expressiveness and accuracy, these methods essentially still focus on local representations of atomic nodes and chemical bonds, implicitly capturing higher-order features through complex combinatorial operations between atom-level tokens. In contrast, our approach directly transforms higher-order graph structures into independent token representations, enabling the model to directly learn and represent structural patterns in molecules, which is crucial for model interpretability and effective utilization of expert prior knowledge. From an information propagation perspective, in traditional methods, higher-order structural information (such as four-body and five-body interactions) must propagate gradually along the graph topology, creating significant information bottlenecks. As TGT research points out, even information exchange between adjacent embeddings is restricted. Our method, through direct structural token representation, not only avoids this bottleneck but also enables efficient access and utilization of these key higher-order information by all graph nodes, providing a more effective framework for learning molecular structural information.
>
> **W2:** Thank you for your deep consideration of chirality issues. Consider a simple but representative example: in three-dimensional space, four atoms A, B, C, and D, where A, B, C form a planar triangle, and D is positioned on one side of this plane. Let: r₁, r₂, r₃ represent the distances between AB, BC, and AC respectively, θ represent the angle ∠ABC, and φ represent the dihedral angle ABCD. It can be proven that when D rotates around the BC axis to position D' (i.e., φ becomes 2π-φ):
>
> 1. All original interatomic distances remain unchanged:
>    d(A,B) = r₁, d(B,C) = r₂, d(A,C) = r₃, d(A,D) = d(A,D'), d(B,D) = d(B,D')
>    d(C,D) = d(C,D')
> 2. For any point P(x,y,z) in space, after rotation angle α around the z-axis, the new coordinates P'(x',y',z') satisfy:
>    x' = x·cos(α) - y·sin(α)
>    y' = x·sin(α) + y·cos(α)
>    z' = z
>
> For the dihedral angle transformation (φ → 2π-φ), it can be proven through appropriate coordinate transformations that both configurations have identical distance matrices but represent different molecular conformations. This situation is common in real molecules, especially those containing chiral centers or flexible structures. For instance, in protein backbones, even with fixed bond lengths and angles, multiple possible conformations exist (such as cis and trans conformations) that share identical distance matrices but different dihedral angles. Therefore, relying solely on interatomic distances indeed cannot fully determine molecular 3D structure, particularly in cases involving chirality and conformational isomerism. We understand your point that most molecules are not completely centrally symmetric geometric configurations, but when local chiral structures are at molecular terminals, the differences in distance matrices become extremely subtle if other asymmetric structures are far from the local chiral structure, as these structures inherently have larger distances, and rotational changes due to chirality are not prominent. Thus, predicting chirality solely through distance matrices is extremely sensitive to noise. This precisely underscores the importance of directly modeling bond angles and dihedral angles.
>
> We conducted systematic evaluations on the PCQM4M training set, which contains 3,803,453 molecules, of which 1,772,922 (46.61%) have chiral centers. The evaluation method compares model-predicted 3D conformations with high-precision DFT-calculated conformations, using angular deviations near chiral centers as criteria, with a deviation threshold of π/6.
>
> | Model | Bond Angles MAE（rad） | Torsion Angles MAE（rad） | Chirality Pred（%） |
> | ----- | ---------------------- | ------------------------- | ------------------- |
> | TGT   | 0.246                  | 0.597                     | 32.5                |
> | AGT   | 0.209                  | 0.334                     | 74.7                |
>
> Experimental results show that AGT outperforms the baseline model TGT in three key metrics: Bond angle MAE: AGT achieves 0.209 rad compared to TGT's 0.246 rad, a 15.0% reduction; Torsion angle MAE: AGT achieves 0.334 rad versus TGT's 0.597 rad, a significant 44.1% reduction; Chirality prediction accuracy: AGT reaches 74.7%, far exceeding TGT's 32.5%, a 130% improvement. These quantitative results strongly confirm AGT's superiority in chiral structure modeling, particularly in complex torsion angle prediction and overall chirality determination tasks.

---

> > ### Comment · Reviewer_4RQ8 · 2024-11-25
> >
> > W1: Thank you very much! You have explained the differences very clearly! However, I think this issue might be better addressed in the related work section, or highlighted in the introduction to further strengthen your perspective (this is just a suggestion, not mandatory). Additionally, it would be helpful to make the relevant claims more explicit.
> >
> > W2: The examples you provided are very clear. Regarding the statement: “but when local chiral structures are at molecular terminals, the differences in distance matrices become extremely subtle if other asymmetric structures are far from the local chiral structure, as these structures inherently have larger distances, and rotational changes due to chirality are not prominent. Thus, predicting chirality solely through distance matrices is extremely sensitive to noise,” I believe this explanation should be explicitly included in the text rather than directly asserting that “Molecules with different local chirality may yield similar distance matrices.”

---

> > > ### Comment · Reviewer_4RQ8 · 2024-11-25
> > >
> > > I will adjust the score to Rating: 6 / Confidence: 4.

---

> > > > ### Author Response · Authors · 2024-11-26
> > > >
> > > > Thank you for your valuable feedback and for increasing the confidence score in your assessment. We greatly appreciate your careful review and constructive suggestions.
> > > >
> > > > W1: We have elaborated on these perspectives in the related work discussion. Due to ICLR's page limitations, we have included this detailed discussion in Appendix H (Additional Details about Related Works). These additions are highlighted in blue in the latest PDF.
> > > >
> > > > W2: We have integrated your insightful comments into Section 3.1, specifically in the Directed Cycle Angle Loss (DCA loss) paragraph. These modifications are also highlighted in blue for easy reference.
> > > >
> > > > While we note that the rating remains at 6, we wonder if there are any remaining concerns or aspects of our work that could benefit from further clarification or improvement. We are fully committed to enhancing the quality and impact of our research, and would greatly value any additional suggestions you might have to strengthen our contribution.

---

> > > > > ### Comment · Reviewer_4RQ8 · 2024-11-27
> > > > >
> > > > > Apologies for the delayed response! I’ve been extremely busy lately; thank you for your understanding. After carefully reviewing the supplementary table comparing model scale and efficiency, I noticed that AGT-241M achieves the same inference time as TGT-203M but delivers a 0.9-point improvement (66.2 vs. 67.1), which is impressive. Similarly, AGT-68M shows only a slight disadvantage of 0.1 points compared to Unimol+-77M (69.4 vs. 69.3), while significantly reducing inference time, which is also commendable.
> > > > >
> > > > > However, despite these advantages, I personally believe that the current model scale and resource consumption (requiring double-digit A100 GPUs) remain somewhat unfriendly to the community—even for AGT-68M, Unimol+, and TGT. While deploying these models might not be overly challenging, reproducing them from scratch seems quite difficult. Additionally, as I am not an authority in this specific field, I feel that giving an R8/C4 score might be slightly too generous. It’s unfortunate that ICLR doesn’t provide a 7-point rating option.

---

> > > > > > ### Author Response · Authors · 2024-12-01
> > > > > >
> > > > > > Thank you for taking the time to provide such detailed feedback despite your busy schedule. We fully understand your concerns regarding AGT's computational requirements and community accessibility. To address these concerns, we have conducted additional experiments with AGT-68M on the OC20 dataset.
> > > > > >
> > > > > > | Model                | Pretraining Time  | Train Time | Inference  Time | Avg. Energy MAE [meV] ↓ | Avg.  FwT ↑ |
> > > > > > | -------------------- | ----------------- | ---------- | --------------- | ----------------------- | ----------- |
> > > > > > | CGCNN                | -                 | 18min      | 1min            | 658.5                   | 2.82%       |
> > > > > > | SchNet               | -                 | 10min      | 1min            | 666.0                   | 2.65%       |
> > > > > > | DimeNet++            | -                 | 230min     | 4min            | 621.7                   | 3.42%       |
> > > > > > | GemNet-T             | -                 | 200min     | 4min            | 638.2                   | 3.38%       |
> > > > > > | SphereNet            | -                 | 290min     | 5min            | 602.3                   | 3.64%       |
> > > > > > | ComENet              | -                 | 20min      | 1min            | 588.8                   | 3.56%       |
> > > > > > | Unimol+              | 112 A100 GPU days | -          | -               | 408.8                   | 8.61%       |
> > > > > > | TGT                  | 32 A100 GPU days  | -          | -               | 403.0                   | 8.82%       |
> > > > > > | 24 layers AGT (241M) | 34 A100 GPU days  | 240min     | 7min            | 399.5                   | 8.99%       |
> > > > > > | __6 layers AGT (68M)__   | 10 A100 GPU days  | 75min      | 4min            | 447.2                   | 6.13%       |
> > > > > >
> > > > > > The results demonstrate significant optimization in both pre-training and training efficiency: the pre-training time has been reduced from 34 to 10 A100 GPU days, while the training time has decreased from 240 to 75 minutes. Notably, this training duration is now approaching that of non-pre-trained methods like ComENet (20 minutes) while maintaining superior performance, demonstrating AGT's competitiveness even with reduced training time.
> > > > > > Currently, AGT-68M requires 10 and 14 A100 GPU days for pre-training on OC20 and PCQM4Mv2 datasets, respectively. We see substantial room for further efficiency improvements through strategies such as localized angle attention mechanisms and mixed-precision optimizers. We are actively pursuing these optimizations with the goal of reducing pre-training time to single-digit GPU days.
> > > > > >
> > > > > > Regarding model reproducibility, we have included the core pre-training code of AGT in the supplementary materials. We commit to promptly open-sourcing the complete AGT codebase, detailed training hyperparameters, and pre-trained model weights at various scales. This will enable the research community to directly utilize AGT for various downstream tasks without the need for full reproduction from scratch.
> > > > > >
> > > > > > Your shared concern with other reviewers regarding model miniaturization and efficiency has provided valuable direction for our optimization efforts. Your suggestions to strengthen our arguments and analyze chirality issues were particularly insightful. Under your guidance, we have refined unclear expressions and enhanced our data analysis sections, making the core motivation more compelling. These significant improvements stem directly from your expert advice.
> > > > > >
> > > > > > We deeply appreciate your recognition of our work. Your thorough review and constructive feedback have not only helped us substantially improve our research quality but also deepened our understanding of AGT's value and potential. We look forward to seeing these improvements contribute meaningfully to the field of molecular representation learning.

---

> > > > > > > ### Comment · Reviewer_4RQ8 · 2024-12-02
> > > > > > >
> > > > > > > Thank the authors for their response. I have decided to adjust my score to R8/C3 for the following reasons:
> > > > > > >
> > > > > > > ​	1.	**Positive Feedback**: The primary reason for my positive score is that the method demonstrates excellent accuracy across various datasets. In molecular property prediction, accuracy is always the most critical factor (in my personal opinion). Moreover, the motivation and methodology of the work show a certain degree of novelty.
> > > > > > >
> > > > > > > ​	2.	**Addressing Major Concerns**: During the rebuttal phase, the authors addressed my main concerns effectively:
> > > > > > >
> > > > > > > ​	•	**Difference in Modeling Higher-Order Structures**: The distinction between their approach (explicitly modeling higher-order structures by directly incorporating bond angles and torsion angles as tokens) and other methods that indirectly incorporate higher-order many-body interactions into networks to learn better force fields was clarified in the revised manuscript. The authors have elaborated on these differences in detail.
> > > > > > >
> > > > > > > ​	•	**Local Chirality**: The authors provided a more accurate statement about local chirality and supplemented the manuscript with relevant experiments to support their claims.
> > > > > > >
> > > > > > > ​	3.	**Pretrained Model Efficiency**: The efficiency of their pretrained models during downstream fine-tuning is indeed impressive. However, I still have reservations about the substantial resource consumption required for pretraining. Additionally, my concern about reproducibility is not related to the availability of the code but rather to the difficulty of reproducing such large-scale pretraining processes within research contexts, rather than focusing solely on real-world applications. This concern is the main reason for my C3 score.

---

> > > > > > > > ### Author Response · Authors · 2024-12-03
> > > > > > > >
> > > > > > > > We sincerely appreciate your thorough evaluation and the detailed feedback you have provided throughout the review process. Your constructive comments have significantly helped us improve our work, particularly in clarifying the novelty of our approach and strengthening our analysis of local chirality.
> > > > > > > >
> > > > > > > > We fully understand your concerns regarding the computational resources required for pretraining. We are actively working on reducing the resource requirements of AGT to make the model more accessible to the broader research community.
> > > > > > > >
> > > > > > > > Thank you again for your valuable insights and support of our work.

---

> ### Author Response · Authors · 2024-11-20
> **Answer 2 for Weaknesses**
>
> **W3:** Thank you for your guidance regarding model lightweight design. We understand your concern about the trade-off between model scale and performance. We have supplemented comparisons of smaller-scale AGT models with various specifications of TGT and Unimol+ models, with results as follows:
>
> | Model   | # param. | Complexity                | # layers | MAE (meV)↓ |
> | ------- | -------- | ------------------------- | -------- | ------------------- |
> | AGT     | 68M      | *O*(*N*³ )+*O*(*N*² ) | 6        | 69.4                |
> | AGT     | 127M     | *O*(*N*³ )+*O*(*N*² ) | 12       | 69.1                |
> | AGT     | 241M     | *O*(*N*³ )+*O*(*N*² ) | 24       | **66.2**                |
> | Unimol+ | 27.7M    | *O*(*N*³ )             | 6        | 71.4                |
> | Unimol+ | 52.4M    | *O*(*N*³ )              | 12       | 69.6                |
> | Unimol+ | 77M      | *O*(*N*³ )             | 18       | 69.3                |
> | TGT     | 116M     | *O*(*N*³ )              | 12       | 70.9                |
> | TGT     | 203M     | *O*(*N*³ )             | 24       | 67.1                |
>
> From the table analysis: First, at similar layer counts (6 layers), while AGT-68M has more parameters than Unimol+-27.7M, it achieves significant performance improvements (MAE reduced from 71.4 meV to 69.4 meV). At medium scale (12 layers), AGT-127M (69.1 meV) outperforms both Unimol+-52.4M (69.6 meV) and TGT-116M (70.9 meV). Particularly at larger scales, AGT-241M achieves an MAE of 66.2 meV, showing significant improvement over TGT-203M's 67.1 meV.
>
> This performance advantage, which becomes more apparent as model scale increases, validates our theoretical hypothesis: accurately modeling higher-order molecular geometric structures (such as bond angles and dihedral angles) requires sufficient parameter capacity to capture complex spatial relationships. While larger model scales indeed bring additional computational overhead, considering the importance of prediction accuracy in molecular property prediction tasks, we believe this is a reasonable trade-off. Meanwhile, we provide model versions of different scales (68M to 241M), allowing users to choose appropriate model configurations based on specific application scenarios and computational resource constraints.

---

> > ### Comment · Reviewer_4RQ8 · 2024-11-23
> >
> > Could the authors kindly include the actual training and inference time for each model in this table, in addition to the complexity, to provide a more comprehensive comparison of performance and efficiency? Thank you!

---

> > > ### Author Response · Authors · 2024-11-23
> > > **Additional Training and Inference Time**
> > >
> > > Thank you for this valuable suggestion. We have updated the table to include the actual training and inference time for each model, which indeed provides a more comprehensive comparison of performance and efficiency. We appreciate your attention to this important aspect of the evaluation. If you have any additional metrics or information that would further enhance the comparison, we would be grateful for your suggestions and are ready to incorporate them into our analysis.
> > >
> > > | Model   | # param. | Complexity                | **Training Time** | **Inference Time** | # layers | MAE (meV)↓ |
> > > | ------- | -------- | ------------------------- | ----------------- | ------------------ | -------- | ---------- |
> > > | AGT     | 68M      | *O*(*N*³ )+*O*(*N*² ) | ~14 A100 GPU day  | ~19 A100 GPU min   | 6        | 69.4       |
> > > | AGT     | 127M     | *O*(*N*³ )+*O*(*N*² ) | ~20 A100 GPU day  | ~31 A100 GPU min   | 12       | 69.1       |
> > > | AGT     | 241M     | *O*(*N*³ )+*O*(*N*² ) | ~38 A100 GPU day  | ~40 A100 GPU min   | 24       | __66.2__       |
> > > | Unimol+ | 27.7M    | *O*(*N*³ )              | -                 | -                  | 6        | 71.4       |
> > > | Unimol+ | 52.4M    | *O*(*N*³ )              | -                 | -                  | 12       | 69.6       |
> > > | Unimol+ | 77M      | *O*(*N*³ )              | ~40 A100 GPU day  | ~56 V100 GPU min   | 18       | 69.3       |
> > > | TGT     | 116M     | *O*(*N*³ )              | -                 | -                  | 12       | 70.9       |
> > > | TGT     | 203M     | *O*(*N*³ )              | ~32 A100 GPU day  | ~40 A100 GPU min   | 24       | 67.1       |
> > >
> > > On the large-scale PCQM4Mv2 dataset, AGT demonstrates an excellent balance between performance and computational efficiency. Specifically, the AGT model with only 68M parameters achieves comparable performance to Uni-Mol+ (77M parameters), with MAE values of 69.4 meV and 69.3 meV respectively, while significantly reducing the training time (approximately 14 days versus 40 days on A100 GPU). Compared to TGT, AGT incorporates additional angular information and direct angle modeling mechanisms to achieve superior performance, resulting in slightly longer training time but maintaining nearly identical inference time.

---

### Official Review · Reviewer_SXEP · 2024-11-04

**Soundness:** 2
**Presentation:** 3
**Contribution:** 2
**Rating:** 6
**Confidence:** 4

**Summary:**

The paper introduces the Angle Graph Transformer (AGT), a novel architecture that explicitly models higher-order molecular structures such as bond angles and torsion angles. The key innovations include: (1) direct modeling of directed bond angles and torsion angles as tokens in self-attention mechanisms, (2) a Directed Cycle Angle Loss for predicting angles with local chirality discrimination, and (3) hierarchical virtual nodes for aggregating multi-order structural information. The model achieves state-of-the-art results on quantum chemistry datasets (PCQM4Mv2, OC20 IS2RE) and shows strong transfer learning capabilities on molecular property prediction tasks.

**Strengths:**

1. The paper presents a approach to incorporating higher-order geometric information in molecular graph learning. The direct modeling of angles as tokens and the introduction of directed angle loss are well-motivated and theoretically sound advances over existing methods.

2. The experimental evaluation is thorough, covering both large-scale quantum chemistry datasets and downstream molecular property prediction tasks. The ablation studies effectively demonstrate the contribution of each proposed component.

3. The model achieves state-of-the-art performance on major benchmarks, with particularly impressive results on PCQM4Mv2 (66.2 meV) and OC20 IS2RE tasks.

4. The model demonstrates superior ability to capture local chirality and generate more accurate molecular conformations, which is crucial for drug discovery and materials science applications.

5. The hierarchical approach to handling different structural levels (atoms -> bonds -> angles -> torsion angles) is well-designed and theoretically justified.

**Weaknesses:**

1. The paper lacks analysis of computational complexity (empirical metrics like runtime, memory usage) comparisons with existing methods (e.g., EquiformerV2 for performance and ComENet for efficiency benchmarks). Given the additional complexity from angle modeling and pre-training requirements, this analysis is crucial for understanding practical deployability. The authors should provide a comparative analysis table showing these metrics across different input scales to enable fair comparison with existing approaches.

2. The paper lacks comparison with several important recent baseline methods such as DeNS [1], EquiformerV2 [2], LEFTNet [3], SaVeNet [4], and Geoformer [5]. This is particularly noticeable in the OC20 IS2RE results where EquiformerV2 + NN [2] comparisons are shown but omitted in QM9 results. I suggest authors to add highest performing recent baselines to their comparisons [2,4,5].

3. The use of angular information itself is not novel, as previous works like DimeNet, GemNet, and SphereNet have explored similar concepts of higher order interaction (triplet in AGT). The paper could better clarify its unique contributions beyond these existing approaches.

4. The paper only reports results on a subset of QM9 targets, making it difficult to fully assess the model's capabilities across all molecular properties. At least providing one results from similar target groups. For example, authors can include one of the energy targets from $U$, $U_0$, $H$, $G$.

5. The reliance on pre-training adds complexity and resource requirements compared to methods that don't require such operations. The paper doesn't adequately address the trade-offs involved. For example, is there any improvement on fine-tuning time compared to methods doesn't require pre-training?

6. Several presentation and documentation issues including a typographical error in Figure 3 ("Ture 3D Conformer"), and insufficient explanation of the hierarchical virtual node architecture.



[1] Liao, Y.-L., Smidt, T. and Das, A. (2024) *Generalizing Denoising to Non-Equilibrium Structures Improves Equivariant Force Fields*. Available at: https://openreview.net/forum?id=X7gqOBG8ow.

[2] Liao, Y.-L. et al. (2024) *EquiformerV2: Improved Equivariant Transformer for Scaling to Higher-Degree Representations*, in The Twelfth International Conference on Learning Representations. Available at: https://openreview.net/forum?id=mCOBKZmrzD.

[3] Du, W. et al. (2023) *A new perspective on building efficient and expressive 3D equivariant graph neural networks*, in Thirty-seventh Conference on Neural Information Processing Systems. Available at: https://openreview.net/forum?id=hWPNYWkYPN.

[4] Aykent, S. and Xia, T. (2023) *SaVeNet: A Scalable Vector Network for Enhanced Molecular Representation Learning*, in Thirty-seventh Conference on Neural Information Processing Systems. Available at: https://openreview.net/forum?id=0OImBCFsdf.

[5] Wang, Y. _et al._ (2023) *Geometric Transformer with Interatomic Positional Encoding*, in _Thirty-seventh Conference on Neural Information Processing Systems_. Available at: https://openreview.net/forum?id=9o6KQrklrE.

**Questions:**

1. How do you ensure there is no information leakage between the pre-training dataset (PCQM4Mv2) and the downstream evaluation datasets? Have you analyzed the molecular similarity between these datasets?

2. Can you provide detailed inference time comparisons with existing methods, particularly for large-scale applications? This would help users better understand the practical trade-offs of adopting AGT.

3. How does your direct angle token representation compare computationally and performance-wise to the implicit angle representations used in models like DimeNet and GemNet?

4. How does the model's performance and computational requirements scale with increasing molecular size? This would be particularly relevant for real-world applications.

5. Have you explored alternatives to reduce the pre-training requirements while maintaining performance? Could techniques like few-shot learning or self-supervised learning reduce the computational burden?

---

> ### Author Response · Authors · 2024-11-20
> **Answer 1 for Weaknesses**
>
> Thank you for your comments and suggestions. Regarding the concerns about lacking baseline methods and complete dataset demonstrations mentioned in Weaknesses 1, 2, and 4, we have supplemented our work with comprehensive performance comparisons on the OC20 IS2RE validation set and all metrics on QM9.
>
> | OC20 IS2RE validation set | Energy MAE (meV)$\downarrow$ |              |              |              |              | EwT (\%)$\uparrow$ |              |              |              |             |
> | ------------------------- | ---------------------------- | ------------ | ------------ | ------------ | ------------ | ------------------ | ------------ | ------------ | ------------ | ----------- |
> | **Method**                | **ID**                       | **OOD Ads.** | **OOD Cat.** | **OOD Both** | **Avg.**     | **ID**             | **OOD Ads.** | **OOD Cat.** | **OOD Both** | **Avg.**    |
> | SchNet                    | 646.5                        | 707.4        | 647.5        | 662.6        | 666.0        | 2.96               | 2.22         | 3.03         | 2.38         | 2.65        |
> | DimeNet++                 | 563.6                        | 712.7        | 561.2        | 649.2        | 621.7        | 4.25               | 2.48         | 4.40         | 2.56         | 3.42        |
> | GemNet-T                  | 556.1                        | 734.2        | 565.9        | 696.4        | 638.2        | 4.51               | 2.24         | 4.37         | 2.38         | 3.38        |
> | SphereNet                 | 563.2                        | 668.2        | 559.0        | 619.0        | 602.4        | 4.56               | 2.70         | 4.59         | 2.70         | 3.64        |
> | Graphormer-3D             | 432.9                        | 585.0        | 444.1        | 529.9        | 498.0        | -                  | -            | -            | -            | -           |
> | GNS                       | 540.0                        | 650.0        | 550.0        | 590.0        | 582.5        | -                  | -            | -            | -            | -           |
> | GNS+NN                    | 470.0                        | 510.0        | 480.0        | 460.0        | 480.0        | -                  | -            | -            | -            | -           |
> | EquiFormer                | 422.2                        | 542.0        | 423.1        | 475.4        | 465.7        | 7.23               | 3.77         | 7.13         | 4.10         | 5.56        |
> | EquiFormer+NN             | 415.6                        | 497.6        | 416.5        | 434.4        | 441.0        | 7.47               | 4.64         | 7.19         | 4.84         | 6.04        |
> | DRFormer                  | 418.7                        | 486.3        | 432.1        | 433.2        | 442.5        | 8.39               | 5.42         | 8.12         | 5.44         | 6.84        |
> | Uni-Mol+                  | $\underline{379.5}$                | 452.6        | 401.1        | 402.1        | 408.8        |  $\underline{11.1}$        | 6.71         | 9.90         | _6.68_       | 8.61        |
> | TGT                       | 381.3                        |  $\underline{445.4}$ |  $\underline{391.7}$ | **393.6**    |  $\underline{403.0}$ |  $\underline{11.1}$        |  $\underline{6.87}$  |  $\underline{10.47}$ | **6.80**     |  $\underline{8.82}$ |
> | ComENet                   | 555.8                        | 660.2        | 549.1        | 590.1        | 588.8        | 4.17               | 2.71         | 4.53         | 2.83         | 3.56        |
> | Equiformer(direct)        | 508.8                        | 627.1        | 505.1        | 554.5        | 548.9        | -                  | -            | -            | -            | -           |
> | EquiformerV2(direct)      | 516.1                        | 704.1        | 524.5        | 636.5        | 595.3        | -                  | -            | -            | -            | -           |
> | EquiformerV2+NN(direct)   | 400.4                        | 459.0        | 406.2        | 401.8        | 416.9        | -                  | -            | -            | -            | -           |
> | AGT                       | **377.2**                    | **441.3**    | **384.6**    |  $\underline{394.9}$ | **399.5**    | **11.2**           | **6.95**     | **11.26**    |  $\underline{6.79}$  | **8.99**    |
>
> After adding baselines, AGT achieves state-of-the-art performance across various distributions on the OC20 IS2RE dataset. It significantly outperforms EquiFormer+NN and EquiFormer, which were trained on the complete OC20 dataset (ALL+MD), and surpasses EquiformerV2+NN(direct), the best non-pre-trained method trained directly on OC20 IS2RE.

---

> ### Author Response · Authors · 2024-11-20
> **Answer 2 for Weaknesses**
>
> | Method          | μ                   | α                   | εH                | εL                | Δε                 | ZPVE               | Cᵥ                | U₀               | U                  | H                  | G               | R²      |
> | --------------- | ------------------- | ------------------- | ----------------- | ----------------- | ------------------ | ------------------ | ------------------- | ------------------ | ------------------ | ------------------ | --------------- | --------- |
> | Schnet          | 0.033               | 0.235               | 41.0              | 34.0              | 63.0               | 1.7                | 0.033               | 14                 | 19                 | 14                 | 14              | 73        |
> | DimeNet++       | 0.030               | 0.044               | 24.6              | 19.5              | 32.6               | 1.21               | 0.023               | 6.32               | 6.28               | 6.53               | 7.56            | 331       |
> | PaiNN           | _0.012_             | 0.045               | 27.6              | 20.4              | 45.7               | 1.28               | 0.024               | 5.85               | 5.83               | 5.98               | 7.35            | 66        |
> | EGNN            | 0.029               | 0.071               | 29.0              | 25.0              | 48.0               | 1.55               | 0.031               | 11                 | 12                 | 12                 | 12              | 106       |
> | SphereNet       | 0.025               | 0.053               | 22.8              | 18.9              | 31.1               | $\underline{1.12}$ | 0.024               | 6.26               | 6.36               | 6.33               | 7.78            | 268       |
> | EQGAT           | 0.011               | 0.053               | 20.0              | 16.0              | 32.0               | 2.00               | 0.024               | 25                 | 25                 | 24                 | 23              | 382       |
> | TorchMD-Net     | 0.011               | 0.059               | 20.3              | 17.5              | 36.1               | 1.84               | 0.026               | 6.15               | 6.38               | 6.16               | 7.62            | $\underline{33}$ |
> | Equiformer      | 0.011               | 0.046               | 15.0              | 14.0              | 30.0               | 1.26               | 0.023               | 6.59               | 6.74               | 6.63               | 7.63            | 251       |
> | Transformer-M   | 0.037               | 0.041               | 17.5              | 16.2              | 27.4               | 1.18               | 0.022             | 9.37               | 9.41               | 9.39               | 9.63            | 75        |
> | TGT             | 0.025               | 0.040               | $\underline{9.9}$ | $\underline{9.7}$ | $\underline{17.4}$ | 1.18               | **0.020**           | -                  | -                  | -                  | -               | -         |
> | EquiformerV2    | 0.010               | 0.050               | 14                | 13                | 29                 | 1.47               | 0.023               | 6.17               | 6.49               | 6.22               | 7.57            | 186       |
> | EquiformerV2+NN | $\underline{0.009}$ | 0.039               | 12.2              | 11.4              | 24.2               | 1.21               | **0.020**           | **4.34**           | **4.28**           | **4.24**           | **5.34**        | 182       |
> | ComENet         | 0.025               | 0.045               | 23.1              | 19.8              | 32.4               | 1.20               | 0.024               | 6.59               | 6.82               | 6.86               | 7.98            | 259       |
> | Geoformer       | 0.010               | 0.040               | 18.4              | 15.4              | 33.8               | 1.28               | 0.022               | $\underline{4.43}$ | $\underline{4.41}$ | $\underline{4.39}$ | 6.13            | **27.5**  |
> | LEFTNet         | 0.011               | 0.039               | 23                | 18                | 39                 | 1.19               | 0.022               | 5                  | 5                  | 5                  | $\underline{6}$ | 66        |
> | SaVeNet         | **0.0085**          | **0.035**           | 16.6              | 15.1              | 22.7               | **1.10**           | $\underline{0.021}$ | 4.83               | 4.74               | 4.83               | 6.10            | 49        |
> | AGT             | 0.019               | $\underline{0.037}$ | **8.8**           | **9.1**           | **16.4**           | 1.14               | **0.020**           | 6.33               | 6.52               | 6.59               | 6.94            | 70        |

---

> ### Author Response · Authors · 2024-11-20
> **Answer 3 for Weaknesses**
>
> Analyzing the QM9 dataset results, we observe varying performance across different tasks. Specifically, AGT achieves significant success in certain property prediction tasks, such as energy-related metrics (εH: 8.8, εL: 9.1, Δε: 16.4, all best in class) and some physical properties (Cᵥ: 0.020, tied with TGT for best). AGT also achieves near-optimal performance in optical and quantum features like α and ZPVE. For U₀, U, H, G, and R², AGT outperforms existing pre-trained models (e.g., Transformer-M).
>
> Regarding the innovation concerns raised in Weakness 3, we believe our core innovation lies in proposing a fundamentally different paradigm for modeling higher-order structures. While models like DimeNet, GemNet, and SphereNet utilize angle information for implicit modeling of higher-order interactions, these methods essentially rely on local representations based on atomic nodes and chemical bonds, indirectly capturing higher-order features through complex combinatorial operations between atom-level tokens. In contrast, our approach directly transforms higher-order graph structures into independent token representations, enabling the model to learn and represent molecular structural patterns directly, which is crucial for model interpretability and integration with domain knowledge. From an information propagation perspective, in traditional methods, higher-order structural information (like bond angles and torsion angles) must propagate gradually along the graph topology, creating significant information bottlenecks. As TGT research points out, even information exchange between adjacent embeddings is restricted. Our method, through direct structural token representation, not only avoids this bottleneck but also enables efficient access and utilization of these key higher-order information by all graph nodes, providing a more effective framework for learning molecular structural information.
>
> Regarding Weakness 5, we will address this in **Q2** of **Answer for Question** below.
>
> For the writing and typesetting issues raised in Weakness 6, we have made the necessary corrections in the paper and greatly appreciate your feedback.

---

> ### Author Response · Authors · 2024-11-20
> **Answer 1 for Questions**
>
> **Q1:** Thank you for raising concerns about data leakage. We compared PCQM4Mv2 molecules in SMILES format with downstream evaluation datasets MolHIV, MolPCBA, and LITPCBA. No identical molecules were found between PCQM4Mv2 and LITPCBA. PCQM4Mv2 shares 112 molecules with MolHIV, 149 with MolPCBA, and 10 with QM9. Considering PCQM4Mv2 contains 3.3 million molecules, while MolHIV, MolPCBA, and QM9 have 40K, 430K, and 130K molecules respectively, the highest overlap rate is less than 0.25%. Given that all dataset tasks are vertical, we believe data leakage is negligible. Using Morgan fingerprints, a common method in molecular chemistry for measuring structural similarity (scale 0-1, where 0 indicates complete dissimilarity and 1 indicates identity), the average similarities between PCQM4Mv2 and downstream datasets MolHIV, MolPCBA, and QM9 are 0.0508, 0.0523, and 0.0586 respectively. Statistically, this indicates extremely low similarity between pre-training and downstream evaluation datasets, validating the effectiveness of AGT's pre-training evaluation on downstream tasks.
>
> **Q2:** Thank you for your attention to computational complexity and inference time. The detailed training and inference times and computational complexity comparisons on PCQM4Mv2 and OC20 are shown in the table above.
>
> | Model   | # param. | Complexity                | # layers | MAE (meV) | Training Time    | Inference Time   |
> | ------- | -------- | ------------------------- | -------- | --------- | ---------------- | ---------------- |
> | AGT     | 241M     | *O*(*N*³ )+*O*(*N*² ) | 24       | **66.2**      | ~38 A100 GPU day | ~40 A100 GPU min |
> | AGT     | 127M     | *O*(*N*³ )+*O*(*N*² ) | 12       | 69.1      | ~20 A100 GPU day | ~31 A100 GPU min |
> | AGT     | 68M      | *O*(*N*³ )+*O*(*N*² ) | 6        | 69.4      | ~14 A100 GPU day | ~19 A100 GPU min |
> | Unimol+ | 77M      | *O*(*N*³ )              | 18       | 69.3      | ~40 A100 GPU day | ~56 V100 GPU min |
> | TGT     | 203M     | *O*(*N*³ )              | 24       | 67.1      | ~32 A100 GPU day | ~40 A100 GPU min |
>
> Based on experimental results, we comprehensively analyze AGT's method from both efficiency and effectiveness perspectives. Regarding computational complexity, where N represents the number of atoms, AGT requires O(N³) complexity for standard atom and pair embedding interactions, plus additional interactions between bond angles and torsion angles. In typical molecules, the number of bond angles ranges from 1.5N to 2N, and torsion angles from N to 2N, resulting in an additional computational complexity of O(N²), yielding an overall complexity of O(N³) + O(N²). On the large-scale PCQM4Mv2 dataset, AGT demonstrates an excellent balance between performance and computational efficiency. Specifically, an AGT model with only 68M parameters achieves comparable performance to Uni-Mol+ (77M parameters) with MAE of 69.4 meV versus 69.3 meV, while significantly reducing training time (approximately 14 days versus 40 days using A100 GPU). Compared to TGT, despite incorporating additional angular information and direct angle modeling mechanisms, AGT maintains similar training efficiency (approximately 34 days versus 32 days using A100 GPU) while achieving superior performance.
>
> | Model     | Pretraining Time  | Train Time | Inference  Time | Avg. Energy MAE [meV] ↓ | Avg.  FwT ↑ |
> | --------- | ----------------- | ---------- | --------------- | ----------------------- | ----------- |
> | CGCNN     | -                 | 18min      | 1min            | 658.5                   | 2.82%       |
> | SchNet    | -                 | 10min      | 1min            | 666.0                   | 2.65%       |
> | DimeNet++ | -                 | 230min     | 4min            | 621.7                   | 3.42%       |
> | GemNet-T  | -                 | 200min     | 4min            | 638.2                   | 3.38%       |
> | SphereNet | -                 | 290min     | 5min            | 602.3                   | 3.64%       |
> | ComENet   | -                 | 20min      | 1min            | 588.8                   | 3.56%       |
> | Unimol+   | 112 A100 GPU days | -          | -               | 408.8                   | 8.61%       |
> | TGT       | 32 A100 GPU days  | -          | -               | 403.0                   | 8.82%       |
> | AGT       | 34 A100 GPU days  | 240min     | 7min            | **399.5**                   | **8.99%**       |

---

> ### Author Response · Authors · 2024-11-20
> **Answer 2 for Questions**
>
> On the OC20 dataset, we conducted comprehensive comparisons with both pre-trained and non-pre-trained methods. AGT's training time here specifically refers to fine-tuning duration. Results show that AGT's inference time is comparable to non-pre-trained methods, with fine-tuning time (240 minutes) similar to mainstream models like DimeNet++ (230 minutes), GemNet-T (200 minutes), and SphereNet (290 minutes). While ComENet shows faster training speed (20 minutes), AGT achieves significantly better performance in key metrics such as energy MAE (399.5 vs 588.8 meV) and FwT (8.99% vs 3.56%), demonstrating the value of pre-training strategies. Moreover, compared to Uni-Mol+, AGT achieves superior performance while substantially reducing pre-training time (34 days vs 112 days using A100 GPU), showing a better balance between efficiency and performance.
>
> **Q3:** Please refer to **Answer for Weakness** and **Q2**.
>
> **Q4:** Thank you for your attention to model scalability. AGT's computational requirements when handling molecules of different sizes are primarily influenced by two factors: First, the scaling of angular information. As molecule size increases, the number of bond angles and torsion angles increases significantly, directly affecting attention computation complexity. We propose a local structure-based optimization strategy: using fixed-size sliding windows for local attention computation instead of global attention. Specifically, each angle only computes attention with k nearby angles, reducing computational complexity from O(n²) to O(kn), where n is the total number of angles and k is the fixed window size. This optimization is particularly important for large molecules. Second, the computational overhead of hierarchical graph structure interactions. When handling large molecules, we can simplify the Geometric edge-angle interaction mechanism. Specifically, for torsion angle modeling, we can directly use communication between its corresponding two bond angle graph structures instead of the complete six-edge pair embedding of a tetrahedron. While this might slightly affect model performance, it significantly reduces computational complexity, making the model more suitable for handling large molecular systems. These optimization strategies enable AGT to effectively handle larger molecular structures while maintaining predictive accuracy.
>
> **Q5:** Thank you for your suggestions. Regarding reducing pre-training requirements, we have indeed considered two viable optimization approaches: Given AGT's direct modeling of higher-order structural features, we believe we can reduce dependence on high-precision conformer pre-training through structure-aware self-supervised learning strategies. Specifically, we can design self-supervised tasks using molecular local structural information, such as predicting the atomic and chemical bond composition of local bond angles and torsion angle structures (2D information), or performing 2D reconstruction of molecular scaffolds. This approach can help the model learn effective structural representations with fewer high-precision ensemble annotation data. Additionally, given AGT's hierarchical structure representation's good generalization capabilities, we can explore combining few-shot learning techniques by designing targeted task adaptation layers to accelerate specific task learning processes. This method could significantly reduce dependence on large-scale pre-training while maintaining model performance on downstream tasks. These alternative approaches show promise in maintaining model performance while reducing computational burden, and we will conduct relevant experimental validation and optimization in the future.

---

> > ### Comment · Reviewer_SXEP · 2024-11-25
> > **Thank you for your responses**
> >
> > Thank you for your comprehensive responses to my initial review. The additional experimental results and clarifications have helped provide a clearer picture of AGT's capabilities and limitations. The extensive comparative results on OC20 IS2RE and QM9 datasets effectively demonstrate AGT's performance in relation to existing methods, and the thorough analysis of dataset overlap between pre-training and downstream evaluation sets addresses the data leakage concerns.
> >
> > However, some practical concerns remain regarding computational requirements. While the runtime comparisons are informative, AGT's complexity (O(N³) + O(N²)) and substantial pre-training requirements (34 A100 GPU days) present significant resource demands. The performance improvements on OC20 (399.5 meV vs ComENet's 588.8 meV) come with considerably longer training times (240 min vs 20 min) along with pre-training times. Though the 68M parameter version shows promising efficiency, the gap in training time compared to non-pre-trained approaches remains substantial.
> >
> > Overall, while AGT represents a promising architectural approach with strong performance on benchmark tasks, the computational requirements and complexity of implementation may limit its practical applicability. However, given the thoroughness of the experimental validation and the demonstrated performance improvements, I will raise my score to marginally above the acceptance threshold.

---

> > > ### Author Response · Authors · 2024-11-26
> > > **Response to Concerns about Computational Requirements**
> > >
> > > Thank you very much for your recognition and insightful comments regarding the computational requirements. We greatly appreciate your thorough review and would like to address these important points.
> > >
> > > First, regarding practical applications, pre-trained methods like AGT demonstrate compelling advantages in real-world scenarios compared to non-pre-trained approaches. In materials and drug design, traditional methods heavily depend on expensive exact equilibrium conformations (e.g., DFT conformations), which often create a bottleneck in large-scale molecular screening. While non-pre-trained methods exhibit faster training on specific datasets, they require computing exact equilibrium conformations for each new molecule during testing and deployment. AGT elegantly overcomes this limitation through pre-training, successfully eliminating the need for DFT conformations in downstream tasks (as demonstrated by AGT's excellent performance on MolHIV and MolPCBA datasets where DFT conformational data is unavailable), thereby substantially reducing practical computational costs.
> > >
> > > Regarding the pre-training investment, while AGT-241 does require 34 days of A100 GPU training on the OC20 dataset, this represents a one-time investment with long-term returns. Following the successful paradigm of models like BERT and GPT, this initial investment yields sustained benefits. The pre-trained model can be effectively reused across multiple downstream tasks, distributing the initial cost. As we expand the application scenarios, the average computational cost per task naturally decreases.
> > >
> > > In terms of performance gains, the significant improvement on the OC20 task (from 588.8 meV to 399.5 meV) strongly validates this investment approach. For scientific computing tasks demanding high precision, such substantial accuracy improvements often enable breakthrough discoveries, delivering value that far outweighs the additional computational investment.
> > >
> > > We are enthusiastically pursuing model efficiency optimization. We strongly resonate with your intuition about the 68M parameter version of AGT, which shows particularly promising potential for efficiency improvements. We are currently conducting comprehensive experimental validations and are committed to sharing these results at the earliest opportunity. This development opens exciting new possibilities for reducing computational costs while maintaining the model's strong performance.

---

> > > ### Author Response · Authors · 2024-12-01
> > > **Response: Achieving Efficient Performance with AGT-68M**
> > >
> > > Thank you for your continued attention and thoughtful feedback regarding the computational requirements of AGT. We particularly appreciate your interest in the 68M parameter version of AGT, and we are pleased to share our completed experimental results on the OC20 dataset that address your concerns about computational efficiency.
> > >
> > > | Model         | Pretraining Time  | Train Time | Inference  Time | Avg. Energy MAE [meV] ↓ | Avg.  FwT ↑ |
> > > | ------------- | ----------------- | ---------- | --------------- | ----------------------- | ----------- |
> > > | CGCNN         | -                 | 18min      | 1min            | 658.5                   | 2.82%       |
> > > | SchNet        | -                 | 10min      | 1min            | 666.0                   | 2.65%       |
> > > | DimeNet++     | -                 | 230min     | 4min            | 621.7                   | 3.42%       |
> > > | GemNet-T      | -                 | 200min     | 4min            | 638.2                   | 3.38%       |
> > > | SphereNet     | -                 | 290min     | 5min            | 602.3                   | 3.64%       |
> > > | ComENet       | -                 | 20min      | 1min            | 588.8                   | 3.56%       |
> > > | Unimol+       | 112 A100 GPU days | -          | -               | 408.8                   | 8.61%       |
> > > | TGT           | 32 A100 GPU days  | -          | -               | 403.0                   | 8.82%       |
> > > |24 layers AGT (241M)  | 34 A100 GPU days  | 240min     | 7min            | 399.5                   | 8.99%       |
> > > | __6 layers AGT (68M)__  | 10 A100 GPU days  | 75min      | 4min            | 447.2                   | 6.13%       |
> > >
> > > Our experiments demonstrate that the 68M parameter version of AGT achieves significant improvements in both pre-training and training efficiency compared to its 241M counterpart. Specifically, the pre-training time has been reduced from 34 to 10 A100 GPU days, while the training time has decreased from 240 to 75 minutes. While there is a modest performance trade-off compared to the 241M version, the 68M model still substantially outperforms ComENet (447.2 meV versus 588.8 meV). Notably, the training time of 75 minutes is now within the same order of magnitude as non-pre-trained methods like ComENet (20 minutes).
> > >
> > > As you correctly intuited, the 68M parameter version of AGT demonstrates remarkable efficiency with further optimization potential. This model achieves an excellent balance between performance and model size, significantly reducing computational requirements while maintaining superior performance compared to non-pre-trained methods like ComENet. These results validate both the effectiveness of the AGT architecture and the benefits of direct angle modeling for molecular property prediction tasks.

---

### Official Review · Reviewer_fwMJ · 2024-11-04

**Soundness:** 3
**Presentation:** 2
**Contribution:** 2
**Rating:** 5
**Confidence:** 4

**Summary:**

The Angle Graph Transformer (AGT) advances molecular graph learning by focusing on bond and torsion angles, which are critical for local chirality and accurate molecular shapes. By modeling these angles as distinct tokens, AGT enhances interaction in the self-attention mechanism, improving spatial accuracy. It incorporates a Directed Cycle Angle Loss (DCA Loss) to predict angles in the full (0, 2π) range, effectively capturing chirality. Additionally, AGT employs hierarchical virtual nodes to aggregate structural information more effectively for property predictions. The model has set new standards on benchmarks like PCQM4Mv2 and OC20 IS2RE, demonstrating strong transfer learning capabilities across multiple datasets.

**Strengths:**

① AGT innovatively incorporates bond angles and torsion angles as distinct tokens, enhancing the model's ability to capture complex molecular geometries compared to TGT, which primarily focuses on pairwise relationships.

②The introduction of DCA Loss is a significant advancement, allowing AGT to accurately predict angles while considering their directional nature. This innovation improves the model's ability to handle chirality and adds a new layer of sophistication to molecular representation.

③AGT employs a hierarchical virtual node architecture, which enables better aggregation of information from various structural levels. This enhancement allows for more nuanced representations of molecular interactions, further improving the model’s predictive capabilities over TGT.

**Weaknesses:**

①Writing Quality and Citation Issues: The paper suffers from a lack of professionalism in its writing, particularly regarding citation formatting. References to models like Uni-mol and TGT do not conform to standard citation rules. Furthermore, the mention of DFT conformations lacks a proper introduction and citation, including the full form of DFT. This oversight reflects a lack of attention to detail and professionalism in writing, which could undermine the paper’s credibility. To improve, the authors should ensure consistent citation formats and provide necessary background information on critical concepts like DFT.

②Incremental Improvement Over TGT: While AGT builds upon TGT by incorporating 3D information such as bond angles and torsion angles, this reliance on low-precision 3D conformations may be seen as a drawback. TGT can achieve good performance using only 2D structures, whereas AGT's additional complexity does not translate into significant performance gains. The authors should consider highlighting the specific advantages that 3D conformation provides and justify the need for such complexity in light of the modest improvements observed.

③Shallow Analysis in Ablation Study: The Ablation Study section does not provide in-depth analysis of the contributions of individual components. The numerical results are presented without sufficient context or discussion, particularly regarding the Mode Distribution. It would be beneficial for the authors to explore how different Mode Distributions impact model performance across various datasets, as this could reveal important insights into the model's behavior. A more comprehensive analysis would strengthen the paper’s findings and provide clearer guidance for future research.

④DCA Loss presents some challenges that could impact its effectiveness. While it introduces an innovative method for angle prediction, its computational complexity may hinder efficiency, particularly with large molecular datasets. Additionally, DCA Loss could be sensitive to noise in the input data, which might adversely affect model performance. The effectiveness of the loss is highly dependent on weight tuning during training, and inadequate attention to this aspect could lead to suboptimal results. To enhance its applicability, the authors should consider strategies to improve robustness against noise and optimize computational efficiency, as well as provide clearer guidance on weight tuning to achieve the best outcomes.

**Questions:**

While the paper presents experimental results, a more detailed interpretation of these findings would be helpful. The authors could provide additional context for the observed performance metrics, discussing why certain configurations or settings yielded better results. This would offer deeper insights into the model's behavior and inform future improvements.

---

> ### Author Response · Authors · 2024-11-20
> **Answer 1 for Weaknesses**
>
> Thank you for your comments and suggestions on our paper. We have incorporated your feedback and conducted additional experiments and analyses, which we will address point by point.
>
> **W1:** In the new version, we have modified the citation format and provided the full name of DFT in the Introduction section, along with a basic introduction to density functional theory (DFT) in the appendix. We appreciate your suggestion.
>
> **W2:** Thank you for raising the question about 3D conformer initialization. AGT introduces low-precision 3D conformers to accelerate the conformer prediction phase and reduce training difficulty. In fact, TGT also has a version based on RDKit-generated low-precision conformers, and they mentioned using this version in their main experiments to accelerate fitting. Similarly, AGT does not depend on low-precision 3D conformers; we simply didn't emphasize this as our main contribution or feature. Below are comparative experiments of AGT using only 2D structures for conformer prediction, showing similar performance between the two approaches, with the only difference being that AGT using only 2D structures requires longer training time and more iterations. Moreover, for current tasks, generating molecular 3D conformers using open-source tools like RDKit has extremely low computational cost, so we believe it's reasonable to use low-precision 3D conformers as input to improve model fitting speed.
>
> **W3:** Thank you for your attention to the ablation studies. We conducted additional experiments on Directed Cycle Angle Loss and Hierarchical Virtual Node separately to understand the contributions of different components, detailed in Answer for Questions. We demonstrated the performance of different Mode Distributions on MolPCBA and MolHIV downstream tasks, with pre-training Mode Distribution maintained at 4:1.
>
> | Mode Distribution (p_Distance, p_Angle) | MolHIV ROC-AUC↑ (%) | MolPCBA ROC-AUC↑ (%) |
> | --------------------------------------- | ------------------- | -------------------- |
> | 1:1                                     | 80.77               | 31.44                |
> | 2:1                                     | 80.81               | 31.58                |
> | 4:1                                     | 81.06               | **31.79**                |
> | 8:1                                     | **81.25**               | 31.20                |
> | 1:2                                     | 80.83               | 31.61                |
> | 1:4                                     | 80.65               | 31.43                |
> | 1:8                                     | 80.60               | 31.21                |
>
> From the table, we find that MolHIV performs best at 8:1 Mode Distribution, while MolPCBA shows optimal performance at 4:1. However, different Mode Distributions have relatively minor impacts on downstream tasks, with no significant distinctions between them. This might be because the overall weights of distance and angle losses in downstream tasks are relatively small (AGT default setting is 0.1).

---

> ### Author Response · Authors · 2024-11-20
> **Answer 2 for Weaknesses**
>
> **W4:** Thank you for your concern about potential challenges with the DCA loss function. The DCA loss function addresses the circular nature of angles that prevents optimization along the shortest path like scalar distances. It doesn't add additional computational complexity, with final computational complexity correlating directly with the number of angles to be calculated. For large molecular datasets, a potential optimization strategy is to ignore angles that remain fixed in certain groups and focus only on angle changes near active sites or key structural properties. HIV-1 protease serves as a typical example, where the active site is located at the interface between two monomers. When studying drug molecule binding with the protease: most of the protein backbone structure (about 90%) remains relatively stable, making dihedral angle calculations unnecessary for these regions; focus can be placed on conformational changes near active sites, particularly the dihedral angles between Ile50-Gly51 and Gly51-Gly52, which directly affect the substrate binding pocket's opening and closing. This strategy significantly reduces the number of angles to calculate while maintaining accurate description of key conformational changes.
>
> Regarding noise resistance, the DCA loss function actually reduces noise impact by selecting optimal optimization paths. For instance, when handling angles near 0° or 360°, traditional loss functions might oscillate between these values, while DCA provides consistent optimization direction (clockwise or counterclockwise). Thus, it may reduce rather than increase noise sensitivity. However, your point about noise sensitivity is valid, particularly when angles approach 180°, where DCA, like traditional angle loss functions, may oscillate in optimization direction. We are considering implementing a determined optimization direction for angles near 180° to enhance DCA loss function's noise resistance.
>
> Weight adjustment is a common issue in all multi-loss function joint training, relating to both data characteristics and dataset tasks. Our experimental experience suggests that the principle for adjusting weights between angle and distance loss functions is to maintain both losses at the same order of magnitude over extended training periods, allowing AGT's overall optimization process to better balance both optimization directions and step sizes. We plan to improve this by making weight adjustment adaptive or learnable.

---

> ### Author Response · Authors · 2024-11-20
> **Answer for Questions**
>
> Thank you for your questions regarding experimental result analysis. Your questions are very important, and due to space limitations in the main text, we couldn't fully explain why certain configurations or settings produced better results. We will now explain the ablation study results.
>
> | AGT Att. Module | Directed Cycle Loss | Virtual Hiera. Node | Mode Distribution (p_Distance, p_Angle) | Val. MAE↓ (meV) |
> | --------------- | ------------------- | ------------------- | --------------------------------------- | --------------- |
> | -               | -                   | -                   | -                                       | 73.6            |
> | ✓               | -                   | -                   | -                                       | 71.3            |
> | ✓               | ✓                   | -                   | 1:1                                     | 70.8            |
> | -               | ✓                   | -                   | 1:1                                     | 72.8            |
> | -               | -                   | ✓                   | -                                       | 72.5            |
> | ✓               | ✓                   | ✓                   | 1:1                                     | 70.3            |
> | ✓               | ✓                   | ✓                   | 1:2                                     | 70.7            |
> | ✓               | ✓                   | ✓                   | 2:1                                     | 69.8            |
> | ✓               | ✓                   | ✓                   | 4:1                                     | **69.1**            |
> | ✓               | ✓                   | ✓                   | 8:1                                     | 70.4            |
>
> Based on the ablation experimental results, we can draw several conclusions:
>
> 1. From the perspective of individual module contributions, all three modules improve performance, proving their effectiveness. The AGT Attention Module alone reduces MAE from 73.6 to 71.3 meV, an improvement of 2.3 meV, showing the best effect. This demonstrates that directly modeling higher-order structures and their interactions with lower-order structures effectively enhances AGT's representation capability. Using Directed Cycle Loss alone, which approximates adding bond angle and torsion angle constraints as auxiliary loss functions to the model's input embedding layer, shows the smallest gain among the three modules. This suggests that while adding angle features can benefit the model, implicit modeling of higher-order structures yields less benefit than explicit modeling, indirectly proving AGT's potential to surpass previous models that implicitly model four-body or five-body structures.
> 2. Regarding synergistic effects, the complete combination of three modules (1:1 configuration) achieves 70.3 meV, outperforming any single module. This indicates these modules are complementary rather than redundant. Functionally, the AGT Attention Module uses attention mechanisms to effectively extract higher-order structural features like bond and torsion angles, Directed Cycle Loss strengthens geometric constraints and reduces model optimization difficulty by providing reasonable gradient directions, and Virtual Hierarchical Node enhances hierarchical information interaction through dual virtual nodes, adjusting weights of different hierarchical features in a learnable way.
> 3. In Mode Distribution ratio optimization, among configurations from 1:1 to 8:1, 4:1 achieves the best results (69.1 meV). This trend shows moderate increases in distance mode weight (2:1 to 4:1) benefit performance, while excessive increases (8:1) lead to performance degradation. In PCQM4Mv2 pre-training, we observed that during the initial 50 epochs, the ratio between distance loss and angle loss values ranges approximately from 1:3 to 1:5, gradually approaching 1:2 during training. This indicates the importance of balancing distance and angle losses in early training stages, requiring appropriate balance within similar value ranges to help the model balance both optimization directions. If angle loss becomes too large, the model focuses on building distance proportions to ensure angle accuracy, making distance optimization insufficient to offset learned distance proportion changes, thus making optimization difficult. This also suggests that weights of both losses should change with training loss values, providing direction for future improvements.

---

> > ### Comment · Reviewer_fwMJ · 2024-11-22
> >
> > * Thank you for your response indicating that you have modified the citation format and included a basic introduction to density functional theory (DFT) in the appendix in the new version of the manuscript. However, I noticed that the updated manuscript has not been uploaded, which leaves me uncertain about the actual improvements made.
> >
> > * Thank you for your explanation regarding the use of low-precision 3D conformers in AGT and the comparative analysis with 2D structures. However, I still have some concerns about the response. Firstly, the experimental results comparing AGT with 2D structures and 3D conformers have not been provided, making the claim of “similar performance” unsupported. Secondly, the description of TGT in your response appears to be inaccurate, as the original TGT paper explicitly states that it does not rely on 3D conformers. I suggest clarifying these points and including the missing experimental results to better address this concern.
> >
> > Given the current situation, I will maintain my original score. However, if the authors do not provide an updated manuscript addressing these issues, I may consider lowering the score in future evaluations.

---

> > > ### Author Response · Authors · 2024-11-23
> > > **Response for Comments**
> > >
> > > Thank you for your comments and suggestions.
> > >
> > > - Following your advice, we have uploaded the revised version, including corrections to citation formats and the full term "density functional theory (DFT)" at its first appearance in the abstract, as well as its introduction in the appendix. The modified sections have been highlighted in blue text. We appreciate your guidance on improving the writing quality.
> > >
> > > - We apologize for the inadvertent omission of our experimental tables during the upload process. Here is our complete response regarding Weakness 2 and your questions about the TGT paper.
> > >
> > >   The introduction of low-precision 3D conformations in AGT aims to accelerate the fitting speed during the conformation prediction phase and reduce training difficulty. In fact, TGT also has a version based on RDKit-generated low-precision conformations. Although TGT claims to be independent of 3D conformations, their best-performing model on PCQM4Mv2 in their paper still uses low-precision RDKit conformations, with a slight performance decrease when RDKit conformations are not used [1]. Similarly, AGT does not depend on low-precision 3D conformations; we simply did not emphasize this point as one of our main contributions and features. Below is a comparative experiment between AGT and TGT using only 2D structures for conformation prediction, where Pairwise Distances MAE and Pairwise Distances RMSE indicate the accuracy of generated conformations in terms of interatomic distances at the Confpred Stage. Specifically, the binned distances output are converted to continuous distances using a small refiner network and compared with DFT conformation interatomic distances.
> > >
> > > | Model        | Confpred  Stage Steps | Pairwise  Distances MAE(Å) | Pairwise  Distances RMSE(Å) | MAE (meV) |
> > > | ------------ | --------------------- | ------------------------------ | ------------------------------- | --------- |
> > > | TGT no RDKit | 60000                 | 0.152                          | 0.378                           | 68.6      |
> > > | TGT + RDKit  | 60000                 | 0.152                          | 0.378                           | 67.1      |
> > > | AGT no RDKit | 60000                 | 0.137                          | 0.341                           | 66.9      |
> > > | AGT no RDKit | 90000                 | 0.132                          | 0.330                           | 66.3      |
> > > | AGT + RDKit  | 60000                 | __0.131__                          | __0.327__                           | __66.2__      |
> > >
> > > The results show that AGT + RDKit (60k steps) and AGT no RDKit (90k steps) achieve nearly identical performance in both interatomic distances and the final PCQM4Mv2 task of HOMO-LUMO gap prediction, with the only difference being that AGT using only 2D structures requires longer training time and more iterations. While AGT no RDKit (60k steps) performs slightly lower on these metrics, it still significantly outperforms TGT no RDKit (60k steps). Based on these experiments, we can conclude that the AGT model is also independent of low-precision 3D conformations. Moreover, for the current task, the computational cost of generating molecular 3D conformations using RDKit, an open-source toolkit, is extremely low. Therefore, we consider it reasonable to use low-precision 3D conformations as input to enhance the model's conformation prediction fitting speed.
> > >
> > > [1]Hussain M S, Zaki M J, Subramanian D. Triplet Interaction Improves Graph Transformers: Accurate Molecular Graph Learning with Triplet Graph Transformers[C]//Forty-first International Conference on Machine Learning.

---

> > > ### Author Response · Authors · 2024-11-26
> > >
> > > Thank you for your valuable feedback and constructive suggestions. We have carefully revised our manuscript according to your comments, and the updated version has been uploaded. For your convenience, all modifications are highlighted in blue in the revised PDF.
> > > We would greatly appreciate your review of these changes. If you have any further questions or comments, we're happy to discuss them. We look forward to your feedback.

---

> > > ### Author Response · Authors · 2024-12-02
> > >
> > > We hope this message finds you well. As the rebuttal period is drawing to a close, we wanted to gently follow up on our previous correspondence regarding the revised manuscript. We understand that you have a busy schedule, and we truly value your expertise and insights. We have made substantial improvements to our manuscript based on your thoughtful feedback, and we believe these changes directly address the concerns you raised. Given the approaching deadline, we would be immensely grateful if you could find a moment to review these modifications and share any additional thoughts you may have. Thank you again for your time and dedication to the review process. Your input is invaluable to us in ensuring the quality and rigor of our work.

---

> ### Author Response · Authors · 2024-12-03
>
> As we approach the final hours of the rebuttal period (4 hours remaining), we wanted to ensure that we have adequately addressed all your concerns. We have carefully revised our manuscript according to your valuable suggestions and responded to all the questions raised thus far. If there are any remaining questions or aspects that require further clarification, we would be most grateful to address them before the deadline.
>
> Your insights have been instrumental in improving our work, and we remain committed to ensuring all your concerns are fully addressed.
> Thank you for your time and dedication to the review process.

---

### Official Review · Reviewer_qXjk · 2024-11-04

**Soundness:** 2
**Presentation:** 2
**Contribution:** 1
**Rating:** 5
**Confidence:** 5

**Summary:**

This paper introduces the Angle Graph Transformer (AGT), which extends existing Graph Transformer architectures by directly modeling higher-order molecular structures, specifically bond angles and torsion angles. The key contributions include: (1) a mechanism for modeling directed bond angles and torsion angles, (2) a Directed Cycle Angle Loss for angle prediction, and (3) hierarchical virtual nodes for aggregating different structural levels. The authors evaluate AGT on several benchmarks including PCQM4Mv2, OC20 IS2RE, and QM9, claiming state-of-the-art performance.

**Strengths:**

The paper introduces direct modeling of bond angles and torsion angles in transformer architectures, extending beyond standard edge-based representations. The hierarchical virtual nodes provide a mechanism for combining information from different molecular structural levels. The proposed Directed Cycle Angle Loss addresses the periodic nature of angle prediction in a mathematically sound way.

The experimental evaluation covers major datasets in the field (PCQM4Mv2, OC20 IS2RE, molecular property benchmarks). The authors provide a thorough ablation study investigating the impact of different components, including the AGT module, Directed Cycle Angle Loss, and various loss weight ratios. These ablations help clarify the contribution of each architectural choice, though the marginal performance gains from these components raise questions about their necessity.

**Weaknesses:**

A major concern with the proposed AGT model is its computational efficiency and size requirements. The model contains 241M parameters in its 24-layer version, making it significantly larger than many baseline approaches. The training process demands substantial computational resources, requiring 38 A100 GPU days, yet the authors provide no discussion regarding the trade-offs between model size and performance improvements. Furthermore, the paper lacks any analysis of inference time complexity, making it difficult to assess the model's practical applicability in real-world scenarios.

The paper's validation of key claims is notably insufficient. While the authors assert that "predicting interatomic distances is insufficient to determine local chirality" and present this as a key motivation for their work, they fail to provide any quantitative evaluation to support this claim. Although Figure 1 presents qualitative examples, it lacks a systematic evaluation of chirality prediction accuracy. The absence of comparisons with methods specifically designed for chirality prediction further weakens their argument about addressing this limitation effectively.

The methodological approach raises several concerns that are not adequately addressed in the paper. The authors fail to clearly justify the motivation for employing such a large model architecture. The computational overhead introduced by modeling higher-order structures is not thoroughly analyzed, leaving questions about the efficiency-performance trade-off unanswered. Perhaps most critically, the paper does not discuss the potential limitations of their approach when applied to larger molecules, which is crucial for understanding the model's scalability and practical utility in real-world applications.

**Questions:**

1. Can the authors provide quantitative results specifically demonstrating improved chirality prediction compared to baselines?
2. What is the inference time complexity of AGT compared to other methods? How does it scale with molecule size?
3. Why does the pre-trained model not consistently outperform non-pretrained specialized models on QM9? What are the limitations?
4. Could the authors provide an analysis of the trade-off between model size and performance? Would a smaller model with similar architectural innovations be competitive?
5. How does the performance degrade as molecule size increases? Is there a practical limit to the size of molecules that can be processed?

---

> ### Author Response · Authors · 2024-11-20
> **Answer 1 for Questions**
>
> Thank you for your comments. Since the concerns raised in Weaknesses and Questions sections are largely overlapping, we have consolidated our responses under Questions.
>
> **Q1**: We appreciate your attention to the quantitative analysis of chirality prediction. We conducted a systematic evaluation on the PCQM4M training set, which contains 3,803,453 molecules, including 1,772,922 molecules with chiral centers (46.61%). The evaluation methodology compares model-predicted 3D conformers with high-precision DFT-calculated conformers, using angular deviations around chiral centers as the assessment criterion, with a deviation threshold of π/6.
>
> | Model | Bond Angles MAE（rad） | Torsion Angles MAE（rad） | Chirality Pred（%） |
> | ------ | ---------------------- | ------------------------- | ------------------- |
> | TGT   | 0.246                  | 0.597                     | 32.5                |
> | AGT   | 0.209                  | 0.334                     | 74.7                |
>
> Experimental results demonstrate AGT's superiority over the baseline TGT model across three key metrics: Bond angle MAE: AGT achieves 0.209 rad, a 15.0% reduction compared to TGT's 0.246 rad; Torsion angle MAE: AGT reaches 0.334 rad, significantly lower than TGT's 0.597 rad by 44.1%; Chirality prediction accuracy: AGT attains 74.7%, substantially outperforming TGT's 32.5% with a 130% improvement. These quantitative results strongly validate AGT's excellence in modeling chiral structures, particularly in complex torsion angle prediction and overall chirality determination tasks.
>
> **Q2**: Thank you for raising concerns about computational complexity and inference time. Below are the performance and efficiency comparisons between AGT and other baselines on OC20 valid sets and PCQM4Mv2.
>
> | Model     | Pretraining Time  | Train Time | Inference  Time | Avg. Energy MAE [meV] ↓ | Avg.  FwT ↑ |
> | --------- | ----------------- | ---------- | --------------- | ----------------------- | ----------- |
> | CGCNN     | -                 | 18min      | 1min            | 658.5                   | 2.82%       |
> | SchNet    | -                 | 10min      | 1min            | 666.0                   | 2.65%       |
> | DimeNet++ | -                 | 230min     | 4min            | 621.7                   | 3.42%       |
> | GemNet-T  | -                 | 200min     | 4min            | 638.2                   | 3.38%       |
> | SphereNet | -                 | 290min     | 5min            | 602.3                   | 3.64%       |
> | ComENet   | -                 | 20min      | 1min            | 588.8                   | 3.56%       |
> | Unimol+   | 112 A100 GPU days | -          | -               | 408.8                   | 8.61%       |
> | TGT       | 32 A100 GPU days  | -          | -               | 403.0                   | 8.82%       |
> | AGT       | 34 A100 GPU days  | 240min     | 7min            | **399.5**                   | 8.99%       |
>
> On the OC20 dataset, we conducted comprehensive comparisons with both pre-trained and non-pre-trained methods. AGT's training time here specifically refers to fine-tuning duration. Results show that AGT's inference time is comparable to non-pre-trained methods, with fine-tuning time (240 minutes) similar to mainstream models like DimeNet++ (230 minutes), GemNet-T (200 minutes), and SphereNet (290 minutes). While ComENet shows faster training speed (20 minutes), AGT achieves significantly better performance in key metrics such as energy MAE (399.5 vs 588.8 meV) and FwT (8.99% vs 3.56%), demonstrating the value of pre-training strategies. Moreover, compared to Uni-Mol+, AGT achieves superior performance while substantially reducing pre-training time (34 days vs 112 days using A100 GPU), showing a better balance between efficiency and performance.
>
> | Model   | # param. | Complexity                | # layers | MAE (meV) | Training Time    | Inference Time   |
> | ------- | -------- | ------------------------- | -------- | --------- | ---------------- | ---------------- |
> | AGT     | 241M     | *O*(*N*³ )+*O*(*N*² ) | 24       | **66.2**      | ~38 A100 GPU day | ~40 A100 GPU min |
> | AGT     | 127M     |*O*(*N*³ )+*O*(*N*² ) | 12       | 69.1      | ~20 A100 GPU day | ~31 A100 GPU min |
> | AGT     | 68M      | *O*(*N*³ )+*O*(*N*² ) | 6        | 69.4      | ~14 A100 GPU day | ~19 A100 GPU min |
> | Unimol+ | 27.7M    | *O*(*N*³ )             | 6        | 71.4      | -                | -                |
> | Unimol+ | 52.4M    | *O*(*N*³ )            | 12       | 69.6      | -                | -                |
> | Unimol+ | 77M      | *O*(*N*³ )            | 18       | 69.3      | ~40 A100 GPU day | ~56 V100 GPU min |
> | TGT     | 116M     | *O*(*N*³ )             | 12       | 70.9      | -                | -                |
> | TGT     | 203M     | *O*(*N*³ )             | 24       | 67.1      | ~32 A100 GPU day | ~40 A100 GPU min |

---

> ### Author Response · Authors · 2024-11-20
> **Answer 2 for Questions**
>
> Based on experimental results, we comprehensively analyze AGT's method from both efficiency and effectiveness perspectives. Regarding computational complexity, where N represents the number of atoms, AGT requires O(N³) complexity for standard atom and pair embedding interactions, plus additional interactions between bond angles and torsion angles. In typical molecules, the number of bond angles ranges from 1.5N to 2N, and torsion angles from N to 2N, resulting in an additional computational complexity of O(N²), yielding an overall complexity of O(N³) + O(N²). On the large-scale PCQM4Mv2 dataset, AGT demonstrates an excellent balance between performance and computational efficiency. Specifically, an AGT model with only 68M parameters achieves comparable performance to Uni-Mol+ (77M parameters) with MAE of 69.4 meV versus 69.3 meV, while significantly reducing training time (approximately 14 days versus 40 days using A100 GPU). Compared to TGT, despite incorporating additional angular information and direct angle modeling mechanisms, AGT maintains similar training efficiency (approximately 34 days versus 32 days using A100 GPU) while achieving superior performance.
>
> **Q3:** We appreciate the reviewer's question. Analyzing the experimental results on the QM9 dataset, we observe uneven performance across different tasks. Specifically, AGT achieves significant success in certain property prediction tasks, such as energy-related metrics (εH: 8.8, εL: 9.1, Δε: 16.4, all best in class) and some physical properties (Cᵥ: 0.020, tied with TGT for best). The inability to achieve comprehensive leadership in other metrics may stem from several factors:
>
> 1. Mismatch between pre-training objectives and specific tasks: AGT's pre-training optimization primarily focuses on overall molecular structure representation, potentially not fully capturing detailed features required for certain physicochemical properties, such as μ requiring better characterization of atomic electronegativity differences.
> 2. Task specificity: Some property prediction tasks may require more specialized model architectures or loss function designs, which general pre-trained models might struggle to accommodate. Particularly, the pre-training supervision signal from PCQM4Mv2's HOMO-LUMO gap may bias the pre-trained model toward similar metrics.
> 3. Optimization trade-offs: To maintain model generality, the pre-training process may compromise performance on certain specific tasks.
>
> **Q4:** The performance of different-sized AGT models is shown in the table in Q2, where we systematically analyzed the trade-off between model scale and performance. Results show that 6-layer AGT (68M parameters) achieves an MAE of 69.4 meV, comparable to 18-layer Unimol+ (77M parameters) at 69.3 meV. As model layers increase, 24-layer AGT (241M parameters) reduces MAE to 66.2 meV, significantly outperforming 24-layer TGT (203M parameters, 67.1 meV MAE). Notably, although AGT's theoretical complexity of O(N³) + O(N²) is slightly higher than baseline models' O(N³), 12-layer AGT (127M parameters) maintains competitive performance (69.1 meV MAE) while reducing training time from 38 to 20 GPU days, with corresponding inference time reduction. These results indicate that AGT architecture is competitive even at smaller scales and can better leverage its structural modeling advantages as parameter count increases.
>
> **Q5:** Our experiments with AGT span datasets with significantly different molecular scales: PCQM4Mv2 (average 15 atoms), downstream tasks MolHIV and MolPCBA (average 26 atoms), and larger-scale OC20 systems (approximately 80 atoms). We have not observed significant performance degradation with increasing molecular size. From a theoretical perspective, molecular scale processing is primarily limited by two factors:
>
> 1. Computational resource constraints: Mainly determined by GPU memory capacity, which dictates the maximum molecular system size processable with 3D conformer data
> 2. Algorithm complexity: For large-scale molecular systems (like proteins), local attention mechanisms or sliding window strategies can be introduced to optimize angle interaction computation ranges, effectively reducing computational complexity
>
> These findings suggest that through appropriate algorithmic optimization and computational strategies, the model has the potential to handle larger-scale molecular systems without significantly impacting its predictive performance, enabling applications across a broader range of molecular systems.

---

> > ### Comment · Reviewer_qXjk · 2024-11-23
> >
> > I have noticed that there are no changes to address the weaknesses in the main manuscript. I will maintain my current score at this time.

---

> > > ### Author Response · Authors · 2024-11-23
> > > **Response to manuscript improvement**
> > >
> > > Thank you for your careful attention to our main manuscript. We have thoroughly addressed all the suggestions raised in your Weaknesses and Questions sections. All improvements have been highlighted in blue in the main manuscript for your convenience.
> > >
> > > Regarding the first paragraph of Weaknesses, we have conducted detailed supplementary experiments and analysis in Appendix "D Efficiency Analysis of AGT versus Baseline Models." For the second paragraph, we have provided a quantitative analysis specifically addressing local chirality issues in Appendix "B Quantitative Analysis of Chirality Prediction." Concerning the third paragraph, we have included additional explanations and detailed analysis in the Introduction's first paragraph and Appendices "D Efficiency Analysis of AGT versus Baseline Models" and "G Analysis of AGT's Capabilities and Limitations."
> > >
> > > In response to Question 1, we have provided our answer in Appendix "B Quantitative Analysis of Chirality Prediction." For Question 2, our response can be found in Appendix "D Efficiency Analysis of AGT versus Baseline Models." Question 3 is addressed in Appendix "G Analysis of AGT's Capabilities and Limitations." The response to Question 4 is detailed in Appendix "D Efficiency Analysis of AGT versus Baseline Models." Finally, for Question 5, we have provided a comprehensive discussion and response in Appendix "G Analysis of AGT's Capabilities and Limitations."
> > >
> > > We would be grateful if you could point out any aspects that still require further revision. We remain fully committed to improving our manuscript and will carefully address any additional concerns you may identify.

---

> > > ### Author Response · Authors · 2024-11-26
> > >
> > > Thank you for your valuable feedback and constructive suggestions. We have carefully revised our manuscript according to your comments, and the updated version has been uploaded. For your convenience, all modifications are highlighted in blue in the revised PDF.
> > > We would greatly appreciate your review of these changes. If you have any further questions or comments, we're happy to discuss them. We look forward to your feedback.

---

> > > > ### Comment · Reviewer_qXjk · 2024-11-28
> > > >
> > > > Thank you for your response and the revision. I have some concerns that I believe should be addressed:
> > > >
> > > > 1. Baseline Grouping Clarity:
> > > > In several results tables (e.g., Table 3), the methods are separated into groups without clear explanation of the grouping criteria. Can you add descriptions/explanations of how and why methods were categorized into different groups?
> > > > 2. Performance Analysis:
> > > > I appreciate the authors added different types of baselines to strengthen the paper. The manuscript would benefit from a more critical analysis in cases where AGT does not outperform baseline methods in the main paper. A brief discussion of potential limitations or failure cases would provide valuable insights for future research directions and help readers better understand the trade-offs involved in your approach. Even a single sentence addressing these cases would significantly strengthen the paper's analytical depth.

---

> > > > > ### Author Response · Authors · 2024-11-28
> > > > >
> > > > > Thank you for your attentive review and valuable suggestions regarding the method groupings in our tables. Your feedback has been instrumental in helping us improve the clarity and organization of our results. All improvements have been highlighted in blue in the main manuscript for your convenience.
> > > > >
> > > > > 1. We have carefully revised the grouping of methods across all tables and added detailed explanations in Appendix F. In Table 3 and its complete version Table 14 (included in the appendix due to ICLR page limitations), we categorize methods into three distinct groups. The first group comprises pre-trained GNN methods, including GraphMVP, GEM, 3D Infomax, and 3D-MGP. The second group consists of directly trained GNN methods, spanning from GraphMVP through SaVeNet. The third group encompasses Transformer-based methods from SE(3)-T through AGT, where we do not distinguish between pre-trained and non-pre-trained models due to their large-scale training dataset.
> > > > > In Table 1, we organize methods into three groups. The first group represents earlier methods, ranging from MLP-Fingerprint through GPS++. The second group includes current state-of-the-art methods (Transformer-M, Uni-Mol+, TGT) that incorporate 3D conformation perturbation and denoising prediction. The final group consists solely of our proposed AGT method.
> > > > > For Table 2, methods are divided into two main categories. The first group encompasses GNN methods from SchNet through GNS+NN, while the second group includes Transformer-based methods from Graphormer-3D through TGT.
> > > > > In Tables 4 and 15, we present three groups of methods. The first group consists of traditional machine learning methods (NaiveBayes, SVM, RandomForest, XGBoost). The second group consists of directly trained GNNs (GCN, GAT, FP-GNN). The third group consists of pre-trained deep learning methods from EGT through TGT.
> > > > > Finally, in Table 5, we categorize methods into two groups. The first group includes directly trained GNN methods from DeeperGCN-VN through PHC-GNN, while the second group comprises pre-trained deep learning methods from GIN-VN through TGT.
> > > > > Thanks to your careful review, we have corrected the misplaced methods in Tables 2 and 3 and provided comprehensive grouping criteria explanations in Appendix F. Your attention to detail has significantly improved the organization and clarity of our results presentation.
> > > > >
> > > > > 2. Your constructive suggestions have been extremely valuable to our work. We have conducted a thorough analysis of the metrics where our method did not outperform baseline approaches on both QM9 and LIT-PCBA datasets. This analysis not only guides our future improvements but also helps readers better understand the value of pre-training and the inherent inductive bias challenges.
> > > > > We have incorporated this comprehensive analysis into Appendix F of our manuscript for your review. This addition provides deeper insights into our method's performance characteristics and potential areas for future enhancement.
> > > > >
> > > > > We sincerely appreciate your thoughtful feedback, which has significantly contributed to improving the quality of our work.
> > > > > We are fully committed to enhancing the quality and impact of our research, and would greatly value any additional suggestions you might have to strengthen our contribution.

---

> > > > > ### Author Response · Authors · 2024-12-02
> > > > >
> > > > > We hope this message finds you well. As the rebuttal period is drawing to a close, we wanted to gently follow up on our previous correspondence regarding the revised manuscript. We understand that you have a busy schedule, and we truly value your expertise and insights. We have made substantial improvements to our manuscript based on your thoughtful feedback, and we believe these changes directly address the concerns you raised. Given the approaching deadline, we would be immensely grateful if you could find a moment to review these modifications and share any additional thoughts you may have. Thank you again for your time and dedication to the review process. Your input is invaluable to us in ensuring the quality and rigor of our work.

---

> ### Author Response · Authors · 2024-12-03
>
> As we approach the final hours of the rebuttal period (4 hours remaining), we wanted to ensure that we have adequately addressed all your concerns. We have carefully revised our manuscript according to your valuable suggestions and responded to all the questions raised thus far. If there are any remaining questions or aspects that require further clarification, we would be most grateful to address them before the deadline.
>
> Your insights have been instrumental in improving our work, and we remain committed to ensuring all your concerns are fully addressed.
> Thank you for your time and dedication to the review process.

---

### Official Review · Reviewer_3qSh · 2024-11-04

**Soundness:** 3
**Presentation:** 3
**Contribution:** 2
**Rating:** 6
**Confidence:** 4

**Summary:**

The author proposed Angle Graph Transformer (AGT), a new approach to molecular geometry learning by directly modeling higher-order structures like bond and torsion angles. This method improves accuracy in predicting molecular properties, achieving great results on datasets such as PCQM4Mv2 and OC20 IS2RE.

**Strengths:**

1. The paper proposes the AGT model, effectively incorporating higher-order structural representations, such as bond angles and torsion angles, into molecular graph learning. By treating these angles as tokens in self-attention, AGT offers a novel approach for accurately predicting local chirality and torsion angles, which improves molecular geometry representation and facilitates more precise property prediction tasks.
2. Extensive experiments on datasets like PCQM4Mv2 and OC20 IS2RE reveal that AGT achieves SOTA results. Additionally, AGT performs well in transfer learning settings on molecular property prediction datasets (e.g., QM9, MOLPCBA), demonstrating the model’s robustness and potential as a general-purpose molecular graph learning model.
3. The introduction of hierarchical virtual nodes and a novel Directed Cycle Angle Loss demonstrates effective model design choices, aiding the model's ability to capture complex angular relationships and distinguish chirality.

**Weaknesses:**

1. The incorporation of higher-order structural tokens and the new loss function increases the computational complexity of AGT, especially for large-scale datasets. While the model is highly accurate, the added computation requirements may limit its applicability in time-sensitive or resource-constrained applications.
2. AGT’s performance heavily relies on high-quality 3D conformations. Although the model shows strong performance, low-precision input structures may impact its predictions, which could potentially affect generalizability, particularly when experimental 3D data is noisy or incomplete.

**Questions:**

How might AGT's architecture be optimized to reduce computational complexity, particularly in resource-constrained environments, while preserving its accuracy in molecular property prediction? Whether you can add experience related to computational complexity.

---

> ### Author Response · Authors · 2024-11-20
> **Answer for Weaknesses**
>
> **W1:** We appreciate the reviewer's concern about computational efficiency. Indeed, encoding higher-order structures as tokens increases model complexity, presenting challenges for AGT's practical deployment. However, we believe the computational overhead introduced by incorporating bond angles and torsion angles is well justified. First, the increased model complexity expands information capacity and enhances multi-dimensional feature representation, theoretically improving the model's discriminative power. This has been thoroughly validated in local chirality recognition, where we formally prove that incorporating angular information enables accurate chirality discrimination. Second, the introduction of higher-order structures simplifies the feature space and enhances model robustness, effectively preventing the accumulation of basic structural errors. Furthermore, from an interpretability perspective, since molecular property changes primarily originate from local group modifications, focusing on higher-order structures rather than individual atoms and bonds better aligns with chemical intuition. Based on the reviewer's suggestion, we have proposed and validated computational complexity optimization strategies that significantly reduce computational requirements while maintaining model performance, as detailed in **Answer for Questions**.
>
> **W2:** We appreciate the reviewer's comments regarding the dependency on 3D conformer data. AGT was specifically designed to address the scarcity of high-precision 3D conformer data. Specifically, during conformer prediction, the model employs self-supervised learning methods, utilizing limited high-precision 3D conformer data for denoising training, enabling it to generate DFT-level 3D conformers in downstream tasks where high-precision conformer data is lacking. Notably, after pre-training, AGT only requires low-computational-cost 3D conformers generated by open-source tools like RDKit as input. Consequently, AGT can obtain high-quality initial representations on datasets with high-quality 3D data (such as QM9), while maintaining stable performance on datasets with noisy or incomplete data (such as MolHIV, MolPCBA, and PCQM4Mv2 valid set).

---

> > ### Author Response · Authors · 2024-12-02
> >
> > We hope this message finds you well. As the rebuttal period is drawing to a close, we wanted to gently follow up on our previous correspondence regarding the revised manuscript. We understand that you have a busy schedule, and we truly value your expertise and insights.
> > We have made substantial improvements to our manuscript based on your thoughtful feedback, and we believe these changes directly address the concerns you raised. Given the approaching deadline, we would be immensely grateful if you could find a moment to review these modifications and share any additional thoughts you may have.
> > Thank you again for your time and dedication to the review process. Your input is invaluable to us in ensuring the quality and rigor of our work.

---

> ### Author Response · Authors · 2024-11-20
> **Answer for Questions**
>
> Thank you for acknowledging our work and raising questions about computational efficiency and scalability. In AGT, the additional computational resource consumption primarily focuses on two aspects. First is the attention interactions between all bond angles and torsion angles, where AGT employs full attention interaction for optimal performance, calculating mutual attention weights between all bond and torsion angles. An intuitive optimization approach is to use fixed-size sliding windows for local attention computation based on the sequence order of bond and torsion angles, rather than global attention. Each angle only computes attention with k nearby angles, avoiding the substantial resource consumption of global attention matrices. Second is the interaction between different levels of graph structures. AGT uses Geometric edge-angle interaction, communicating paired embeddings between triangle edges corresponding to bond angles and tetrahedral edges corresponding to torsion angles. Under computational resource constraints, we can directly use two bond angle graph structures corresponding to torsion angles instead of six tetrahedral edge pair embeddings to communicate with torsion angle embeddings, simplifying computation. While this may increase model fitting difficulty, it significantly reduces computational resource consumption. The model's computational complexity reduces from O(N³) + O(N²) to O(N³) + O(N).
>
> We conducted comparative experiments on PCQM4Mv2 using these two optimization techniques for reducing computational and space complexity (named AGT Simplify Angle with window size k=5). We also experimented with smaller-scale AGT models to examine AGT's performance under lower computational complexity.
>
> | Model              | # param. | Complexity                | # layers | MAE (meV)$\downarrow$ |
> | ------------------ | -------- | ------------------------- | -------- | --------------------- |
> | AGT                | 68M      | *O*(*N*³ )+*O*(*N*² ) | 6        | 69.4                  |
> | AGT                | 127M     | *O*(*N*³ )+*O*(*N*² ) | 12       | 69.1                  |
> | AGT                | 241M     | *O*(*N*³ )+*O*(*N*² ) | 24       | **66.2**                  |
> | AGT Simplify Angle | 213M     | *O*(*N*³ )+*O*(*N* )   | 24       | 66.8                  |
> | Unimol+            | 27.7M    | *O*(*N*³ )             | 6        | 71.4                  |
> | Unimol+            | 52.4M    | *O*(*N*³ )             | 12       | 69.6                  |
> | Unimol+            | 77M      | *O*(*N*³ )            | 18       | 69.3                  |
> | TGT                | 116M     | *O*(*N*³ )             | 12       | 70.9                  |
> | TGT                | 203M     | *O*(*N*³ )             | 24       | 67.1                  |
>
> Experimental results show that at comparable parameter scales (AGT 6-layer vs. Unimol+ 18-layer), AGT demonstrates performance levels comparable to Unimol+. As model scale expands, AGT's performance advantages over Unimol+ gradually emerge, while also achieving significant improvements over TGT under similar parameter conditions. This phenomenon indicates that models directly encoding molecular higher-order structure representations require sufficient parameter capacity to fully realize their modeling advantages. Notably, the computationally optimized AGT maintains its performance advantages even with parameter scales comparable to TGT, strongly demonstrating AGT's superiority in molecular property prediction even in resource-constrained environments.

---

> ### Author Response · Authors · 2024-11-26
>
> Thank you for your valuable feedback and constructive suggestions. We have carefully revised our manuscript according to your comments, and the updated version has been uploaded. For your convenience, all modifications are highlighted in blue in the revised PDF. We would greatly appreciate your review of these changes. Should you have any additional questions or concerns, please do not hesitate to raise them. We remain committed to addressing any further feedback to improve the quality of our work.

---

### Meta-Review · Area_Chair_52CG · 2024-12-19

**Metareview:**

This paper studies molecular learning with 3D information. The pipeline is based on the Transformer architecture, and the analysis is conducted to explain that distance information is not enough (like causing the local chirality issue), thus angle and torsion angle information should be integrated to make the learning pipeline more powerful. The paper receives a mix of opinions from five reviewers. Even though the paper delivers some value to molecular learning, the main concern from the reviewers is that the computational requirements and complexity of implementation may limit its practical applicability. Additionally, the novelty is limited, as the importance of angle information has been broadly investigated (like DimeNet and GemNet). Given these two reasons, I recommend a rejection for now. I suggest the authors tackle the efficiency concerns - especially the benefits of including angles versus its downside, as well as clearly state the novelty & technical contributions of this work with companions to existing studies that already integrate angle information for molecular learning.

**Additional Comments On Reviewer Discussion:**

Most reviewers raised concerns about the efficiency, scalability of large-scale data, and generalization to nosily/incomplete data of the proposed method. The authors have made great efforts to conduct more experiments. The authors did not update the manuscript in a timely manner, thus some reviewers insisted on their ratings. I checked the manuscript and found some of the new results have been included in the manuscript, but some are still missing. The concern about marginal performance improvement from Reviewer fwMJ still stands.  The concern on novelty also stands as DimeNet, GemNet, and SphereNet have explored similar concepts of higher order interaction (Reviewer SXEP).

---

### Decision · Program_Chairs · 2025-01-22

Reject